# Bootstrap Model Aggregation for Distributed Statistical Learning

**Jun Han**
Department of Computer Science
Dartmouth College
jun.han.gr@dartmouth.edu

**Qiang Liu**
Department of Computer Science
Dartmouth College
qiang.liu@dartmouth.edu

## Abstract

In distributed, or privacy-preserving learning, we are often given a set of probabilistic models estimated from different local repositories, and asked to combine them into a single model that gives efficient statistical estimation. A simple method is to linearly average the parameters of the local models, which, however, tends to be degenerate or not applicable on non-convex models, or models with different parameter dimensions. One more practical strategy is to generate bootstrap samples from the local models, and then learn a joint model based on the combined bootstrap set. Unfortunately, the bootstrap procedure introduces additional noise and can significantly deteriorate the performance. In this work, we propose two variance reduction methods to correct the bootstrap noise, including a weighted M-estimator that is both statistically efficient and practically powerful. Both theoretical and empirical analysis is provided to demonstrate our methods.

## 1 Introduction

Modern data science applications increasingly involve learning complex probabilistic models over massive datasets. In many cases, the datasets are distributed into multiple machines at different locations, between which communication is expensive or restricted; this can be either because the data volume is too large to store or process in a single machine, or due to privacy constraints as these in healthcare or financial systems. There has been a recent growing interest in developing *communication-efficient* algorithms for probabilistic learning with distributed datasets; see e.g., Boyd et al. (2011); Zhang et al. (2012); Dekel et al. (2012); Liu and Ihler (2014); Rosenblatt and Nadler (2014) and reference therein.

This work focuses on a *one-shot* approach for distributed learning, in which we first learn a set of local models from local machines, and then combine them in a fusion center to form a single model that integrates all the information in the local models. This approach is highly efficient in both computation and communication costs, but casts a challenge in designing statistically efficient combination strategies. Many studies have been focused on a simple *linear averaging* method that linearly averages the parameters of the local models (e.g., Zhang et al., 2012, 2013; Rosenblatt and Nadler, 2014); although nearly optimal asymptotic error rates can be achieved, this simple method tends to degenerate in practical scenarios for models with non-convex log-likelihood or non-identifiable parameters (such as latent variable models, and neural models), and is not applicable at all for models with non-additive parameters (e.g., when the parameters have discrete or categorical values, or the parameter dimensions of the local models are different).

A better strategy that overcomes all these practical limitations of linear averaging is the *KL-averaging* method (Liu and Ihler, 2014; Merugu and Ghosh, 2003), which finds a model that minimizes the sum of Kullback-Leibler (KL) divergence to all the local models. In this way, we directly combine the models, instead of the parameters. The exact *KL-averaging* is not computationally tractable

because of the intractability of calculating KL divergence; a practical approach is to draw (bootstrap) samples from the given local models, and then learn a combined model based on all the bootstrap data. Unfortunately, the bootstrap noise can easily dominate in this approach and we need a very large bootstrap sample size to obtain accurate results. In Section 3, we show that the MSE of the estimator obtained from the naive way is $O(N^{-1} + (dn)^{-1})$, where $N$ is the total size of the observed data, and $n$ is bootstrap sample size of each local model and $d$ is the number of machines. This means that to ensure a MSE of $O(N^{-1})$, which is guaranteed by the centralized method and the simple linear averaging, we need $dn \gtrsim N$; this is unsatisfying since $N$ is usually very large by assumption.

In this work, we use variance reduction techniques to cancel out the bootstrap noise and get better KL-averaging estimates. The difficulty of this task is first illustrated using a relatively straightforward control variates method, which unfortunately suffers some of the practical drawback of the linear averaging method due to the use of a linear correction term. We then propose a better method based on a weighted M-estimator, which inherits all the practical advantages of *KL-averaging*. On the theoretical part, we show that our methods give a MSE of $O(N^{-1} + (dn^2)^{-1})$, which significantly improves over the original bootstrap estimator. Empirical studies are provided to verify our theoretical results and demonstrate the practical advantages of our methods.

This paper is organized as follows. Section 2 introduces the background, and Section 3 introduces our methods and analyze their theoretical properties. We present numerical results in Section 4 and conclude the paper in Section 5. Detailed proofs can be found in the appendix.

## 2 Background and Problem Setting

Suppose we have a dataset $X = \{\boldsymbol{x}_j, \ j = 1, 2, ..., N\}$ of size $N$, *i.i.d.* drawn from a probabilistic model $p(\boldsymbol{x}|\boldsymbol{\theta}^*)$ within a parametric family $\mathcal{P} = \{p(\boldsymbol{x}|\boldsymbol{\theta}) : \boldsymbol{\theta} \in \Theta\}$; here $\boldsymbol{\theta}^*$ is the unknown true parameter that we want to estimate based on $X$. In the distributed setting, the dataset $X$ is partitioned into $d$ disjoint subsets, $X = \bigcup_{k=1}^{d} X^k$, where $X^k$ denotes the $k$-th subset which we assume is stored in a local machine. For simplicity, we assume all the subsets have the same data size ($N/d$).

The traditional maximum likelihood estimator (MLE) provides a natural way for estimating the true parameter $\boldsymbol{\theta}^*$ based on the whole dataset $X$,

$$\text{Global MLE:} \quad \hat{\boldsymbol{\theta}}_{\text{mle}} = \underset{\boldsymbol{\theta} \in \Theta}{\arg\max} \sum_{k=1}^{d} \sum_{j=1}^{N/d} \log p(\boldsymbol{x}_j^k \mid \boldsymbol{\theta}), \quad \text{where } X^k = \{\boldsymbol{x}_j^k\}. \tag{1}$$

However, directly calculating the global MLE is challenging due to the distributed partition of the dataset. Although distributed optimization algorithms exist (e.g., Boyd et al., 2011; Shamir et al., 2014), they require iterative communication between the local machines and a fusion center, which can be very time consuming in distributed settings, for which the number of communication rounds forms the main bottleneck (regardless of the amount of information communicated at each round).

We instead consider a simpler *one-shot* approach that first learns a set of local models based on each subset, and then send them to a fusion center in which they are combined into a global model that captures all the information. We assume each of the local models is estimated using a MLE based on subset $X^k$ from the $k$-th machine:

$$\text{Local MLE:} \quad \hat{\boldsymbol{\theta}}_k = \underset{\boldsymbol{\theta} \in \Theta}{\arg\max} \sum_{j=1}^{N/d} \log p(\boldsymbol{x}_j^k \mid \boldsymbol{\theta}), \quad \text{where } k \in [d] = \{1, 2, \cdots, d\}. \tag{2}$$

The major problem is how to combine these local models into a global model. The simplest way is to linearly average all local MLE parameters:

$$\text{Linear Average:} \quad \hat{\boldsymbol{\theta}}_{\text{linear}} = \frac{1}{d} \sum_{k=1}^{d} \hat{\boldsymbol{\theta}}_k.$$

Comprehensive theoretical analysis has been done for $\hat{\boldsymbol{\theta}}_{\text{linear}}$ (e.g., Zhang et al., 2012; Rosenblatt and Nadler, 2014), which show that it has an asymptotic MSE of $\mathbb{E}||\hat{\boldsymbol{\theta}}_{\text{linear}} - \boldsymbol{\theta}^*||^2 = O(N^{-1})$. In fact, it is equivalent to the global MLE $\hat{\boldsymbol{\theta}}_{\text{mle}}$ up to the first order $O(N^{-1})$, and several improvements have been developed to improve the second order term (e.g., Zhang et al., 2012; Huang and Huo, 2015).

Unfortunately, the linear averaging method can easily break down in practice, or is even not applicable when the underlying model is complex. For example, it may work poorly when the likelihood has multiple modes, or when there exist non-identifiable parameters for which different parameter values correspond to a same model (also known as the label-switching problem); models of this kind include latent variable models and neural networks, and appear widely in machine learning. In addition, the linear averaging method is obviously not applicable when the local models have different numbers of parameters (e.g., Gaussian mixtures with unknown numbers of components), or when the parameters are simply not additive (such as parameters with discrete or categorical values). Further discussions on the practical limitations of the linear averaging method can be found in Liu and Ihler (2014).

All these problems of linear averaging can be well addressed by a *KL-averaging* method which averages the model (instead of the parameters) by finding a geometric center of the local models in terms of KL divergence (Merugu and Ghosh, 2003; Liu and Ihler, 2014). Specifically, it finds a model $p(\boldsymbol{x} \mid \boldsymbol{\theta}_{\mathrm{KL}}^*)$ where $\boldsymbol{\theta}_{\mathrm{KL}}^*$ is obtained by $\boldsymbol{\theta}_{\mathrm{KL}}^* = \arg\min_{\boldsymbol{\theta}} \sum_{k=1}^d \mathrm{KL}(p(\boldsymbol{x}|\hat{\boldsymbol{\theta}}_k) \parallel p(\boldsymbol{x}|\boldsymbol{\theta}))$, which is equivalent to,

$$\text{Exact KL Estimator:} \quad \boldsymbol{\theta}_{\mathrm{KL}}^* = \arg\max_{\boldsymbol{\theta} \in \Theta} \left\{ \eta(\boldsymbol{\theta}) \equiv \sum_{k=1}^d \int p(\boldsymbol{x} \mid \hat{\boldsymbol{\theta}}_k) \log p(\boldsymbol{x} \mid \boldsymbol{\theta}) d\boldsymbol{x} \right\}. \quad (3)$$

Liu and Ihler (2014) studied the theoretical properties of the KL-averaging method, and showed that it exactly recovers the global MLE, that is, $\boldsymbol{\theta}_{\mathrm{KL}}^* = \hat{\boldsymbol{\theta}}_{\mathrm{mle}}$, when the distribution family is a full exponential family, and achieves an optimal asymptotic error rate (up to the second order) among all the possible combination methods of $\{\hat{\boldsymbol{\theta}}_k\}$.

Despite the attractive properties, the exact KL-averaging is not computationally tractable except for very simple models. Liu and Ihler (2014) suggested a naive *bootstrap* method for approximation: it draws *parametric bootstrap* sample $\{\widetilde{\boldsymbol{x}}_j^k\}_{j=1}^n$ from each local model $p(\boldsymbol{x}|\hat{\boldsymbol{\theta}}_k)$, $k \in [d]$ and use it to approximate each integral in (3). The optimization in (3) then reduces to a tractable one,

$$\text{KL-Naive Estimator:} \quad \hat{\boldsymbol{\theta}}_{\mathrm{KL}} = \arg\max_{\boldsymbol{\theta} \in \Theta} \left\{ \hat{\eta}(\boldsymbol{\theta}) \equiv \frac{1}{n} \sum_{k=1}^d \sum_{j=1}^n \log p(\widetilde{\boldsymbol{x}}_j^k \mid \boldsymbol{\theta}) \right\}. \quad (4)$$

Intuitively, we can treat each $\widetilde{X}_k = \{\widetilde{\boldsymbol{x}}_j^k\}_{j=1}^n$ as an approximation of the original subset $X^k = \{\boldsymbol{x}_j^k\}_{j=1}^{N/d}$, and hence can be used to approximate the global MLE in (1).

Unfortunately, as we show in the sequel, the accuracy of $\hat{\boldsymbol{\theta}}_{\mathrm{KL}}$ critically depends on the bootstrap sample size $n$, and one would need $n$ to be nearly as large as the original data size $N/d$ to make $\hat{\boldsymbol{\theta}}_{\mathrm{KL}}$ achieve the baseline asymptotic rate $O(N^{-1})$ that the simple linear averaging achieves; this is highly undesirably since $N$ is often assumed to be large in distributed learning settings.

## 3 Main Results

We propose two variance reduction techniques for improving the KL-averaging estimates and discuss their theoretical and practical properties. We start with a concrete analysis on the KL-naive estimator $\hat{\boldsymbol{\theta}}_{\mathrm{KL}}$, which was missing in Liu and Ihler (2014).

**Assumption 1.** *1.* $\log p(\boldsymbol{x} \mid \boldsymbol{\theta})$, $\frac{\partial \log p(\boldsymbol{x}|\boldsymbol{\theta})}{\partial \boldsymbol{\theta}}$, *and* $\frac{\partial^2 \log p(\boldsymbol{x}|\boldsymbol{\theta})}{\partial \boldsymbol{\theta} \partial \boldsymbol{\theta}^\top}$ *are continuous for* $\forall \boldsymbol{x} \in \mathcal{X}$ *and* $\forall \boldsymbol{\theta} \in \Theta$; *2.* $\frac{\partial^2 \log p(\boldsymbol{x}|\boldsymbol{\theta})}{\partial \boldsymbol{\theta} \partial \boldsymbol{\theta}^\top}$ *is positive definite and* $C_1 \leq \|\frac{\partial^2 \log p(\boldsymbol{x}|\boldsymbol{\theta})}{\partial \boldsymbol{\theta} \partial \boldsymbol{\theta}^\top}\| \leq C_2$ *in a neighbor of* $\boldsymbol{\theta}^*$ *for* $\forall x \in \mathcal{X}$, *and* $C_1, C_2$ *are some positive constans.*

**Theorem 2.** *Under Assumption 1,* $\hat{\boldsymbol{\theta}}_{\mathrm{KL}}$ *is a consistent estimator of* $\boldsymbol{\theta}_{\mathrm{KL}}^*$ *as* $n \to \infty$, *and*

$$\mathbb{E}(\hat{\boldsymbol{\theta}}_{\mathrm{KL}} - \boldsymbol{\theta}_{\mathrm{KL}}^*) = o(\frac{1}{dn}), \quad \mathbb{E}\|\hat{\boldsymbol{\theta}}_{\mathrm{KL}} - \boldsymbol{\theta}_{\mathrm{KL}}^*\|^2 = O(\frac{1}{dn}),$$

*where $d$ is the number of machines and $n$ is the bootstrap sample size for each local model $p(\boldsymbol{x} \mid \hat{\boldsymbol{\theta}}_k)$.*

The proof is in Appendix A. Because the MSE between the exact KL estimator $\boldsymbol{\theta}_{\mathrm{KL}}^*$ and the true parameter $\boldsymbol{\theta}^*$ is $O(N^{-1})$ as shown in Liu and Ihler (2014), the MSE between $\hat{\boldsymbol{\theta}}_{\mathrm{KL}}$ and the true

parameter $\boldsymbol{\theta}^*$ is

$$\mathbb{E}\|\hat{\boldsymbol{\theta}}_{\mathrm{KL}} - \boldsymbol{\theta}^*\|^2 \approx \mathbb{E}\|\hat{\boldsymbol{\theta}}_{\mathrm{KL}} - \boldsymbol{\theta}^*_{\mathrm{KL}}\|^2 + \mathbb{E}\|\boldsymbol{\theta}^*_{\mathrm{KL}} - \boldsymbol{\theta}^*\|^2 = O(N^{-1} + (dn)^{-1}). \qquad (5)$$

To make the MSE between $\hat{\boldsymbol{\theta}}_{\mathrm{KL}}$ and $\boldsymbol{\theta}^*$ equal $O(N^{-1})$, as what is achieved by the simple linear averaging, we need draw $dn \gtrsim N$ bootstrap data points in total, which is undesirable since $N$ is often assumed to be very large by the assumption of distributed learning setting (one exception is when the data is distributed due to privacy constraint, in which case $N$ may be relatively small).

Therefore, it is a critical task to develop more accurate methods that can reduce the noise introduced by the bootstrap process. In the sequel, we introduce two variance reduction techniques to achieve this goal. One is based a (linear) control variates method that improves $\hat{\boldsymbol{\theta}}_{\mathrm{KL}}$ using a linear correction term, and another is a *multiplicative* control variates method that modifies the M-estimator in (4) by assigning each bootstrap data point with a positive weight to cancel the noise. We show that both method achieves a higher $O(N^{-1} + (dn^2)^{-1})$ rate under mild assumptions, while the second method has more attractive practical advantages.

### 3.1 Control Variates Estimator

The control variates method is a technique for variance reduction on Monte Carlo estimation (e.g., Wilson, 1984). It introduces a set of correlated auxiliary random variables with known expectations or asymptotics (referred as the control variates), to balance the variation of the original estimator. In our case, since each bootstrapped subsample $\widetilde{X}^k = \{\widetilde{\boldsymbol{x}}_j^k\}_{j=1}^n$ is know to be drawn from the local model $p(\boldsymbol{x} \mid \hat{\boldsymbol{\theta}}_k)$, we can construct a control variate by re-estimating the local model based on $\widetilde{X}^k$:

$$\text{Bootstrapped Local MLE:} \quad \widetilde{\boldsymbol{\theta}}_k = \arg\max_{\boldsymbol{\theta} \in \Theta} \sum_{j=1}^n \log p(\widetilde{\boldsymbol{x}}_j^k \mid \boldsymbol{\theta}), \quad \text{for } k \in [d], \qquad (6)$$

where $\widetilde{\boldsymbol{\theta}}_k$ is known to converge to $\hat{\boldsymbol{\theta}}_k$ asymptotically. This allows us to define the following control variates estimator:

$$\text{KL-Control Estimator:} \quad \hat{\boldsymbol{\theta}}_{\mathrm{KL}-C} = \hat{\boldsymbol{\theta}}_{\mathrm{KL}} + \sum_{k=1}^d \mathfrak{B}_k (\widetilde{\boldsymbol{\theta}}_k - \hat{\boldsymbol{\theta}}_k), \qquad (7)$$

where $\mathfrak{B}_k$ is a matrix chosen to minimize the asymptotic variance of $\hat{\boldsymbol{\theta}}_{\mathrm{KL}-C}$; our derivation shows that the asymptotically optimal $\mathfrak{B}_k$ has a form of

$$\mathfrak{B}_k = -(\sum_{k=1}^d I(\hat{\boldsymbol{\theta}}_k))^{-1} I(\hat{\boldsymbol{\theta}}_k), \quad k \in [d], \qquad (8)$$

where $I(\hat{\boldsymbol{\theta}}_k)$ is the empirical Fisher information matrix of the local model $p(\boldsymbol{x} \mid \hat{\boldsymbol{\theta}}_k)$. Note that this differentiates our method from the typical control variates methods where $\mathfrak{B}_k$ is instead estimated using empirical covariance between the control variates and the original estimator (in our case, we can not directly estimate the covariance because $\hat{\boldsymbol{\theta}}_{\mathrm{KL}}$ and $\widetilde{\boldsymbol{\theta}}_k$ are not averages of i.i.d. samples). The procedure of our method is summarized in Algorithm 1. Note that the form of (7) shares some similarity with the one-step estimator in Huang and Huo (2015), but Huang and Huo (2015) focuses on improving the linear averaging estimator, and is different from our setting.

We analyze the asymptotic property of the estimator $\hat{\boldsymbol{\theta}}_{\mathrm{KL}-C}$, and summarize it as follows.

**Theorem 3.** *Under Assumption (1), $\hat{\boldsymbol{\theta}}_{\mathrm{KL}-C}$ is a consistent estimator of $\boldsymbol{\theta}^*_{\mathrm{KL}}$ as $n \to \infty$, and its asymptotic MSE is guaranteed to be smaller than the KL-naive estimator $\hat{\boldsymbol{\theta}}_{\mathrm{KL}}$, that is,*

$$n\mathbb{E}\|\hat{\boldsymbol{\theta}}_{\mathrm{KL}-C} - \boldsymbol{\theta}^*_{\mathrm{KL}}\|^2 < n\mathbb{E}\|\hat{\boldsymbol{\theta}}_{\mathrm{KL}} - \boldsymbol{\theta}^*_{\mathrm{KL}}\|^2, \quad \text{as } n \to \infty.$$

*In addition, when $N > n \times d$, the $\hat{\boldsymbol{\theta}}_{\mathrm{KL}-C}$ has "zero-variance" in that $\mathbb{E}\|\hat{\boldsymbol{\theta}}_{\mathrm{KL}} - \boldsymbol{\theta}^*_{\mathrm{KL}}\|^2 = O((dn^2)^{-1})$. Further, in terms of estimating the true parameter, we have*

$$\mathbb{E}\|\hat{\boldsymbol{\theta}}_{\mathrm{KL}-C} - \boldsymbol{\theta}^*\|^2 = O(N^{-1} + (dn^2)^{-1}). \qquad (9)$$

**Algorithm 1** KL-Control Variates Method for Combining Local Models

1: **Input:** Local model parameters $\{\hat{\boldsymbol{\theta}}_k\}_{k=1}^d$.
2: Generate bootstrap data $\{\widetilde{\boldsymbol{x}}_j^k\}_{j=1}^n$ from each $p(\boldsymbol{x}|\hat{\boldsymbol{\theta}}_k)$, for $k \in [d]$.
3: Calculate the KL-Naive estimator, $\hat{\boldsymbol{\theta}}_{\mathrm{KL}} = \arg\max_{\boldsymbol{\theta} \in \Theta} \sum_{k=1}^d \frac{1}{n} \sum_{j=1}^n \log p(\widetilde{\boldsymbol{x}}_j^k|\boldsymbol{\theta})$.
4: Re-estimate the local parameters $\widetilde{\boldsymbol{\theta}}_k$ via (6) based on the bootstrapped data subset $\{\widetilde{\boldsymbol{x}}_j^k\}_{j=1}^n$, for $k \in [d]$.
5: Estimate the empirical Fisher information matrix $I(\hat{\boldsymbol{\theta}}_k) = \frac{1}{n} \sum_{j=1}^n \frac{\partial \log p(\widetilde{\boldsymbol{x}}_j^k|\hat{\boldsymbol{\theta}}_k)}{\partial \boldsymbol{\theta}} \frac{\partial \log p(\widetilde{\boldsymbol{x}}_j^k|\hat{\boldsymbol{\theta}}_k)}{\partial \boldsymbol{\theta}}^\top$, for $k \in [d]$.
6: **Ouput:** The parameter $\hat{\boldsymbol{\theta}}_{\mathrm{KL}-C}$ of the combined model is given by (7) and (8).

---

The proof is in Appendix B. From (9), we can see that the MSE between $\hat{\boldsymbol{\theta}}_{\mathrm{KL}-C}$ and $\boldsymbol{\theta}^*$ reduces to $O(N^{-1})$ as long as $n \gtrsim (N/d)^{1/2}$, which is a significant improvement over the KL-naive method which requires $n \gtrsim N/d$. When the goal is to achieve an $O(\epsilon)$ MSE, we would just need to take $n \gtrsim 1/(d\epsilon)^{1/2}$ when $N > 1/\epsilon$, that is, $n$ does not need to increase with $N$ when $N$ is very large.

Meanwhile, because $\hat{\boldsymbol{\theta}}_{\mathrm{KL}-C}$ requires a linear combination of $\hat{\boldsymbol{\theta}}_k$, $\widetilde{\boldsymbol{\theta}}_k$ and $\hat{\boldsymbol{\theta}}_{\mathrm{KL}}$, it carries the practical drawbacks of the linear averaging estimator as we discuss in Section 2. This motivates us to develop another *KL-weighted* method shown in the next section, which achieves the same asymptotical efficiency as $\hat{\boldsymbol{\theta}}_{\mathrm{KL}-C}$, while still inherits all the practical advantages of *KL-averaging*.

## 3.2  KL-Weighted Estimator

Our KL-weighted estimator is based on directly modifying the M-estimator for $\hat{\boldsymbol{\theta}}_{\mathrm{KL}}$ in (4), by assigning each bootstrap data point $\widetilde{\boldsymbol{x}}_j^k$ a positive weight according to the probability ratio $p(\widetilde{\boldsymbol{x}}_j^k \mid \hat{\boldsymbol{\theta}}_k)/p(\widetilde{\boldsymbol{x}}_j^k \mid \widetilde{\boldsymbol{\theta}}_k)$ of the actual local model $p(x|\hat{\boldsymbol{\theta}}_k)$ and the re-estimated model $p(x|\widetilde{\boldsymbol{\theta}}_k)$ in (6). Here the probability ratio acts like a *multiplicative* control variate (Nelson, 1987), which has the advantage of being positive and applicable to non-identifiable, non-additive parameters. Our estimator is defined as

$$\text{KL-Weighted Estimator:} \quad \hat{\boldsymbol{\theta}}_{\mathrm{KL}-W} = \arg\max_{\boldsymbol{\theta} \in \Theta} \left\{ \widetilde{\eta}(\boldsymbol{\theta}) \equiv \sum_{k=1}^d \frac{1}{n} \sum_{j=1}^n \frac{p(\widetilde{\boldsymbol{x}}_j^k|\hat{\boldsymbol{\theta}}_k)}{p(\widetilde{\boldsymbol{x}}_j^k|\widetilde{\boldsymbol{\theta}}_k)} \log p(\widetilde{\boldsymbol{x}}_j^k|\boldsymbol{\theta}) \right\}. \tag{10}$$

We first show that this weighted estimator $\widetilde{\eta}(\boldsymbol{\theta})$ gives a more accurate estimation of $\eta(\boldsymbol{\theta})$ in (3) than the straightforward estimator $\hat{\eta}(\boldsymbol{\theta})$ defined in (4) for any $\boldsymbol{\theta} \in \Theta$.

**Lemma 4.** *As $n \to \infty$, $\widetilde{\eta}(\boldsymbol{\theta})$ is a more accurate estimator of $\eta(\boldsymbol{\theta})$ than $\hat{\eta}(\boldsymbol{\theta})$, in that*

$$n\mathrm{Var}(\widetilde{\eta}(\boldsymbol{\theta})) \leq n\mathrm{Var}(\hat{\eta}(\boldsymbol{\theta})), \quad as \ n \to \infty, \quad for \ any \ \boldsymbol{\theta} \in \Theta. \tag{11}$$

This estimator is motivated by Henmi et al. (2007) in which the same idea is applied to reduce the asymptotic variance in importance sampling. Similar result is also found in Hirano et al. (2003), in which it is shown that a similar weighted estimator with estimated propensity score is more efficient than the estimator using true propensity score in estimating the average treatment effects. Although being a very powerful tool, results of this type seem to be not widely known in machine learning, except several applications in semi-supervised learning (Sokolovska et al., 2008; Kawakita and Kanamori, 2013), and off-policy learning (Li et al., 2015).

We go a step further to analyze the asymptotic property of our weighted M-estimator $\hat{\boldsymbol{\theta}}_{\mathrm{KL}-W}$ that maximizes $\widetilde{\eta}(\boldsymbol{\theta})$. It is natural to expect that the asymptotic variance of $\hat{\boldsymbol{\theta}}_{\mathrm{KL}-W}$ is smaller than that of $\hat{\boldsymbol{\theta}}_{\mathrm{KL}}$ based on maximizing $\hat{\eta}(\boldsymbol{\theta})$; this is shown in the following theorem.

**Theorem 5.** *Under Assumption 1, $\hat{\boldsymbol{\theta}}_{\mathrm{KL}-W}$ is a consistent estimator of $\boldsymbol{\theta}_{\mathrm{KL}}^*$ as $n \to \infty$, and has a better asymptotic variance than $\hat{\boldsymbol{\theta}}_{\mathrm{KL}}$, that is,*

$$n\mathbb{E}\|\hat{\boldsymbol{\theta}}_{\mathrm{KL}-W} - \boldsymbol{\theta}_{\mathrm{KL}}^*\|^2 \leq n\mathbb{E}\|\hat{\boldsymbol{\theta}}_{\mathrm{KL}} - \boldsymbol{\theta}_{\mathrm{KL}}^*\|^2, \quad when \ n \to \infty.$$

**Algorithm 2** KL-Weighted Method for Combining Local Models
---
1: **Input:** Local MLEs $\{\hat{\boldsymbol{\theta}}_k\}_{k=1}^d$.
2: Generate bootstrap sample $\{\widetilde{\boldsymbol{x}}_j^k\}_{j=1}^n$ from each $p(\boldsymbol{x}|\hat{\boldsymbol{\theta}}_k)$, for $k \in [d]$.
3: Re-estimate the local model parameter $\widetilde{\boldsymbol{\theta}}_k$ in (6) based on bootstrap subsample $\{\widetilde{\boldsymbol{x}}_j^k\}_{j=1}^n$, for each $k \in [d]$.
4: **Output:** The parameter $\hat{\boldsymbol{\theta}}_{\text{KL}-W}$ of the combined model is given by (10).
---

*When $N > n \times d$, we have $\mathbb{E}\|\hat{\boldsymbol{\theta}}_{\text{KL}-W} - \boldsymbol{\theta}_{\text{KL}}^*\|^2 = O((dn^2)^{-1})$ as $n \to \infty$. Further, its MSE for estimating the true parameter $\boldsymbol{\theta}^*$ is*

$$\mathbb{E}\|\hat{\boldsymbol{\theta}}_{\text{KL}-W} - \boldsymbol{\theta}^*\|^2 = O(N^{-1} + (dn^2)^{-1}). \tag{12}$$

The proof is in Appendix C. This result is parallel to Theorem 3 for the linear control variates estimator $\hat{\boldsymbol{\theta}}_{\text{KL}-C}$. Similarly, it reduces to an $O(N^{-1})$ rate once we take $n \gtrsim (N/d)^{1/2}$.

Meanwhile, unlike the linear control variates estimator, $\hat{\boldsymbol{\theta}}_{\text{KL}-W}$ inherits all the practical advantages of KL-averaging: it can be applied whenever the KL-naive estimator can be applied, including for models with non-identifiable parameters, or with different numbers of parameters. The implementation of $\hat{\boldsymbol{\theta}}_{\text{KL}-W}$ is also much more convenient (see Algorithm 2), since it does not need to calculate the Fisher information matrix as required by Algorithm 1.

## 4 Empirical Experiments

We study the empirical performance of our methods on both simulated and real world datasets. We first numerically verify the convergence rates predicted by our theoretical results using simulated data, and then demonstrate the effectiveness of our methods in a challenging setting when the number of parameters of the local models are different as decided by Bayesian information criterion (BIC). Finally, we conclude our experiments by testing our methods on a set of real world datasets.

The models we tested include probabilistic principal components analysis (PPCA), mixture of PPCA and Gaussian Mixtures Models (GMM). GMM is given by $p(\boldsymbol{x} \mid \boldsymbol{\theta}) = \sum_{s=1}^m \alpha_s \mathcal{N}(\boldsymbol{\mu}_s, \Sigma_s)$ where $\boldsymbol{\theta} = (\alpha_s, \boldsymbol{\mu}_s, \Sigma_s)$. PPCA model is defined with the help of a hidden variable $\boldsymbol{t}$, $p(\boldsymbol{x} \mid \boldsymbol{\theta}) = \int p(\boldsymbol{x} \mid \boldsymbol{t}; \boldsymbol{\theta})p(\boldsymbol{t} \mid \boldsymbol{\theta})d\boldsymbol{t}$, where $p(\boldsymbol{x} \mid \boldsymbol{t}; \boldsymbol{\theta}) = \mathcal{N}(\boldsymbol{x}; \boldsymbol{\mu} + W\boldsymbol{t}, \sigma^2)$, and $p(\boldsymbol{t} \mid \boldsymbol{\theta}) = \mathcal{N}(\boldsymbol{t}; \boldsymbol{0}, \boldsymbol{I})$ and $\boldsymbol{\theta} = \{\boldsymbol{\mu}, W, \sigma^2\}$. The mixture of PPCA is $p(\boldsymbol{x} \mid \boldsymbol{\theta}) = \sum_{s=1}^m \alpha_s p_s(\boldsymbol{x} \mid \boldsymbol{\theta}_s)$, where $\boldsymbol{\theta} = \{\alpha_s, \boldsymbol{\theta}_s\}_{s=1}^m$ and each $p_s(\boldsymbol{x} \mid \boldsymbol{\theta}_s)$ is a PPCA model.

Because all these models are latent variable models with unidentifiable parameters, the direct linear averaging method are not applicable. For GMM, it is still possible to use a *matched linear averaging* which matches the mixture components of the different local models by minimizing a symmetric KL divergence; the same idea can be used on our linear control variates method to make it applicable to GMM. On the other hand, because the parameters of PPCA-based models are unidentifiable up to arbitrary orthonormal transforms, linear averaging and linear control variates can no longer be applied easily. We use expectation maximization (EM) to learn the parameters in all these three models.

### 4.1 Numerical Verification of the Convergence Rates

We start with verifying the convergence rates in (5), (9) and (12) of MSE $\mathbb{E}\|\hat{\boldsymbol{\theta}} - \boldsymbol{\theta}^*\|^2$ of the different estimators for estimating the true parameters. Because there is also an non-identifiability problem in calculating the MSE, we again use the symmetric KL divergence to match the mixture components, and evaluate the MSE on $WW^\top$ to avoid the non-identifiability w.r.t. orthonormal transforms. To verify the convergence rates w.r.t. $n$, we fix $d$ and let the total dataset $N$ be very large so that $N^{-1}$ is negligible. Figure 1 shows the results when we vary $n$, where we can see that the MSE of KL-naive $\hat{\boldsymbol{\theta}}_{\text{KL}}$ is $O(n^{-1})$ while that of KL-control $\hat{\boldsymbol{\theta}}_{\text{KL}-C}$ and KL-weighted $\hat{\boldsymbol{\theta}}_{\text{KL}-W}$ are $O(n^{-2})$; both are consistent with our results in (5), (9) and (12).

In Figure 2(a), we increase the number $d$ of local machines, while using a fix $n$ and a very large $N$, and find that both $\hat{\boldsymbol{\theta}}_{\text{KL}}$ and $\hat{\boldsymbol{\theta}}_{\text{KL}-W}$ scales as $O(d^{-1})$ as expected. Note that since the total

observation data size $N$ is fixed, the number of data in each local machine is $(N/d)$ and it decreases as we increase $d$. It is interesting to see that the performance of the KL-based methods actually increases with more partitions; this is, of course, with a cost of increasing the total bootstrap sample size $dn$ as $d$ increases. Figure 2(b) considers a different setting, in which we increase $d$ when fixing the total observation data size $N$, and the total bootstrap sample size $n_{\text{tot}} = n \times d$. According to (5) and (12), the MSEs of $\hat{\boldsymbol{\theta}}_{\text{KL}}$ and $\hat{\boldsymbol{\theta}}_{\text{KL}-W}$ should be about $O(n_{\text{tot}}^{-1})$ and $O(dn_{\text{tot}}^{-2})$ respectively when $N$ is very large, and this is consistent with the results in Figure 2(b). It is interesting to note that the MSE of $\hat{\boldsymbol{\theta}}_{\text{KL}}$ is independent with $d$ while that of $\hat{\boldsymbol{\theta}}_{\text{KL}-W}$ increases linearly with $d$. This is not conflict with the fact that $\hat{\boldsymbol{\theta}}_{\text{KL}-W}$ is better than $\hat{\boldsymbol{\theta}}_{\text{KL}}$, since we always have $d \leq n_{\text{tot}}$.

Figure 2(c) shows the result when we set $n = (N/d)^\alpha$ and vary $\alpha$, where we find that $\hat{\boldsymbol{\theta}}_{\text{KL}-W}$ quickly converges to the global MLE as $\alpha$ increases, while the KL-naive estimator $\hat{\boldsymbol{\theta}}_{\text{KL}}$ converges significantly slower. Figure 2(d) demonstrates the case when we increase $N$ while fix $d$ and $n$, where we see our KL-weighted estimator $\hat{\boldsymbol{\theta}}_{\text{KL}-W}$ matches closely with $N$, except when $N$ is very large in which case the $O((dn^2)^{-1})$ term starts to dominate, while KL-naive is much worse. We also find the linear averaging estimator performs poorly, and does not scale with $O(N^{-1})$ as the theoretical rate claims; this is due to unidentifiable orthonormal transform in the PPCA model that we test on.

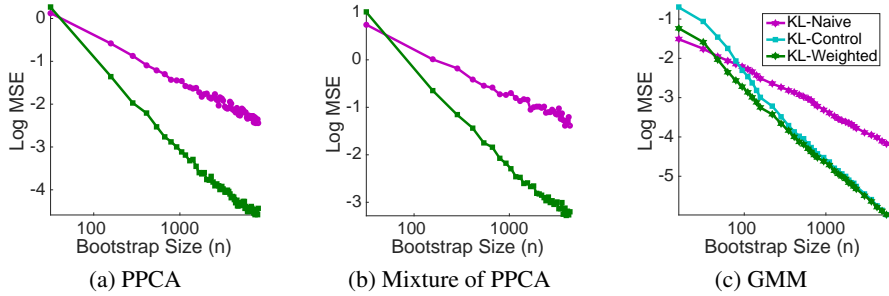

|  |  |  |
|:---:|:---:|:---:|
| (a) PPCA | (b) Mixture of PPCA | (c) GMM |

Figure 1: Results on different models with simulated data when we change the bootstrap sample size $n$, with fixed $d = 10$ and $N = 6 \times 10^7$. The dimensions of the PPCA models in (a)-(b) are 5, and that of GMM in (c) is 3. The numbers of mixture components in (b)-(c) are 3. Linear averaging and KL-Control are not applicable for the PPCA-based models, and are not shown in (a) and (b).

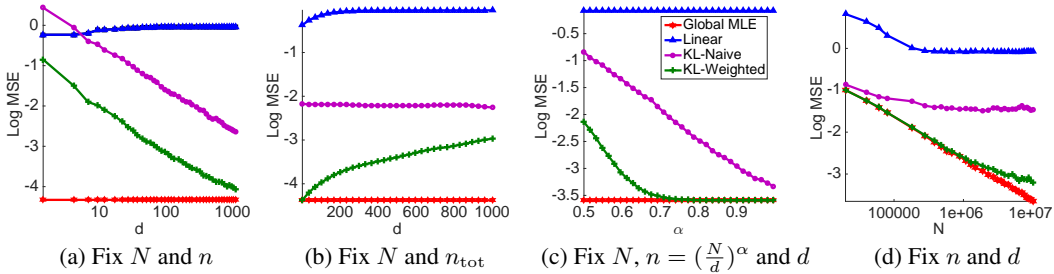

|  |  |  |  |
|:---:|:---:|:---:|:---:|
| (a) Fix $N$ and $n$ | (b) Fix $N$ and $n_{\text{tot}}$ | (c) Fix $N$, $n = (\frac{N}{d})^\alpha$ and $d$ | (d) Fix $n$ and $d$ |

Figure 2: Further experiments on PPCA with simulated data. (a) varying $n$ with fixed $N = 5 \times 10^7$. (b) varying $d$ with $N = 5 \times 10^7$, $n_{\text{tot}} = n \times d = 3 \times 10^5$. (c) varying $\alpha$ with $n = (\frac{N}{d})^\alpha$, $N = 10^7$ and $d$. (d) varying $N$ with $n = 10^3$ and $d = 20$. The dimension of data $\boldsymbol{x}$ is 5 and the dimension of latent variables $\boldsymbol{t}$ is 4.

## 4.2 Gaussian Mixture with Unknown Number of Components

We further apply our methods to a more challenging setting for distributed learning of GMM when the number of mixture components is unknown. In this case, we first learn each local model with EM and decide its number of components using BIC selection. Both linear averaging and KL-control $\hat{\boldsymbol{\theta}}_{\text{KL}-C}$ are not applicable in this setting, and and we only test KL-naive $\hat{\boldsymbol{\theta}}_{\text{KL}}$ and KL-weighted $\hat{\boldsymbol{\theta}}_{\text{KL}-W}$. Since the MSE is also not computable due to the different dimensions, we evaluate $\hat{\boldsymbol{\theta}}_{\text{KL}}$ and $\hat{\boldsymbol{\theta}}_{\text{KL}-W}$ using the log-likelihood on a hold-out testing dataset as shown in Figure 3. We can see that $\hat{\boldsymbol{\theta}}_{\text{KL}-W}$ generally outperforms $\hat{\boldsymbol{\theta}}_{\text{KL}}$ as we expect, and the relative improvement increases

significantly as the dimension of the observation data $x$ increases. This suggests that our variance reduction technique works very efficiently in high dimension problems.

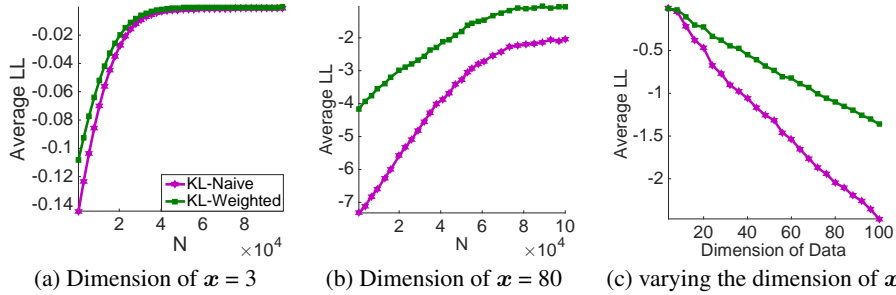

(a) Dimension of $x = 3$   (b) Dimension of $x = 80$   (c) varying the dimension of $x$

Figure 3: GMM with the number of mixture components estimated by BIC. We set $n = 600$ and the true number of mixtures to be 10 in all the cases. (a)-(b) vary the total data size $N$ when the dimension of $x$ is 3 and 80, respectively. (c) varies the dimension of the data with fixed $N = 10^5$. The y-axis is the testing $\log$ likelihood compared with that of global MLE.

## 4.3 Results on Real World Datasets

Finally, we apply our methods to several real world datasets, including the SensIT Vehicle dataset on which mixture of PPCA is tested, and the Covertype and Epsilon datasets on which GMM is tested. From Figure 4, we can see that our KL-Weight and KL-Control (when it is applicable) again perform the best. The (matched) linear averaging performs poorly on GMM (Figure 4(b)-(c)), while is not applicable on mixture of PPCA.

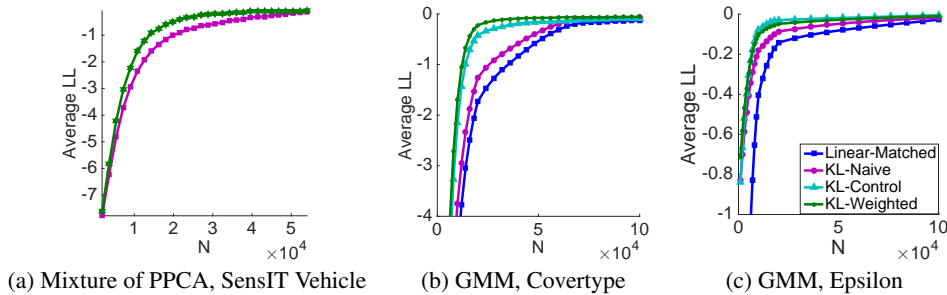

(a) Mixture of PPCA, SensIT Vehicle   (b) GMM, Covertype   (c) GMM, Epsilon

Figure 4: Testing $\log$ likelihood (compared with that of global MLE) on real world datasets. (a) Learning Mixture of PPCA on SensIT Vehicle. (b)-(c) Learning GMM on Covertype and Epsilon. The number of local machines is 10 in all the cases, and the number of mixture components are taken to be the number of labels in the datasets. The dimension of latent variables in (a) is 90. For Epsilon, a PCA is first applied and the top 100 principal components are chosen. Linear-matched and KL-Control are not applicable on Mixture of PPCA and are not shown on (a).

## 5   Conclusion and Discussion

We propose two variance reduction techniques for distributed learning of complex probabilistic models, including a KL-weighted estimator that is both statistically efficient and widely applicable for even challenging practical scenarios. Both theoretical and empirical analysis is provided to demonstrate our methods. Future directions include extending our methods to discriminant learning tasks, as well as the more challenging deep generative networks on which the exact MLE is not computable tractable, and surrogate likelihood methods with stochastic gradient descent are need. We note that the same KL-averaging problem also appears in the "knowledge distillation" problem in Bayesian deep neural networks (Korattikara et al., 2015), and it seems that our technique can be applied straightforwardly.

**Acknowledgement** This work is supported in part by NSF CRII 1565796.

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
