[Supplementary Material]

# Appendix for "Bootstrap Model Aggregation for Distributed Statistical Learning"

**Jun Han**
Department of Computer Science
Dartmouth College
jun.han.gr@dartmouth.edu

**Qiang Liu**
Department of Computer Science
Dartmouth College
qiang.liu@dartmouth.edu

## 1 Appendix A

We study the asymptotic property of the KL-naive estimator $\hat{\boldsymbol{\theta}}_{\mathrm{KL}}$, and prove Theorem 2.

### 1.1 Notations and Assumptions

To simplify the notations for the proofs in the following, we define the following notations.

$$s(\boldsymbol{x};\boldsymbol{\theta}) = \log p(\boldsymbol{x} \mid \boldsymbol{\theta}); \quad \dot{s}(\boldsymbol{x};\boldsymbol{\theta}) = \frac{\partial \log p(\boldsymbol{x} \mid \boldsymbol{\theta})}{\partial \boldsymbol{\theta}}; \quad \ddot{s}(\boldsymbol{x};\boldsymbol{\theta}) = \frac{\partial^2 \log p(\boldsymbol{x} \mid \boldsymbol{\theta})}{\partial \boldsymbol{\theta}^2};$$

$$I(\boldsymbol{\theta}) = \mathbb{E}(\ddot{s}(x,\boldsymbol{\theta})); \quad I(\hat{\boldsymbol{\theta}}_k, \boldsymbol{\theta}^*_{\mathrm{KL}}) = \mathbb{E}(\ddot{s}(\boldsymbol{x}, \boldsymbol{\theta}^*_{\mathrm{KL}}) \mid \hat{\boldsymbol{\theta}}_k). \tag{1}$$

We start with investigating the theoretical property of $\hat{\boldsymbol{\theta}}_{\mathrm{KL}}$.

**Lemma 1.** *Based on Assumption* 1*, as $n \to \infty$, we have $\mathbb{E}(\hat{\boldsymbol{\theta}}_{\mathrm{KL}} - \boldsymbol{\theta}^*_{\mathrm{KL}}) = o((dn)^{-1})$. Further, in terms of estimating the true parameter, we have*

$$\mathbb{E}\|\hat{\boldsymbol{\theta}}_{\mathrm{KL}} - \boldsymbol{\theta}^*\|^2 = O(N^{-1} + (dn)^{-1}). \tag{2}$$

**Proof:** Based on Equation (3) and (4), we know

$$\sum_{k=1}^{d} \frac{1}{n} \sum_{j=1}^{n} \dot{s}(\widetilde{\boldsymbol{x}}_j^k; \hat{\boldsymbol{\theta}}_{\mathrm{KL}}) - \sum_{k=1}^{d} \int p(x|\hat{\boldsymbol{\theta}}_k)\dot{s}(\boldsymbol{x}; \boldsymbol{\theta}^*_{\mathrm{KL}})d\boldsymbol{x} = 0. \tag{3}$$

By the law of large numbers, we can rewrite Equation (3) as

$$\sum_{k=1}^{d} \int p(\boldsymbol{x}|\hat{\boldsymbol{\theta}}_k)\dot{s}(\boldsymbol{x}; \hat{\boldsymbol{\theta}}_{\mathrm{KL}})d\boldsymbol{x} - \sum_{k=1}^{d} \int p(x|\hat{\boldsymbol{\theta}}_k)\dot{s}(\boldsymbol{x}; \boldsymbol{\theta}^*_{\mathrm{KL}})d\boldsymbol{x} = o_p(\frac{1}{n}). \tag{4}$$

We also observe that $\dot{s}(\boldsymbol{x}; \hat{\boldsymbol{\theta}}_{\mathrm{KL}}) - \dot{s}(\boldsymbol{x}; \boldsymbol{\theta}^*_{\mathrm{KL}}) = \left[ \int_0^1 \ddot{s}(\boldsymbol{x}; \boldsymbol{\theta}^*_{\mathrm{KL}} + t(\hat{\boldsymbol{\theta}}_{\mathrm{KL}} - \boldsymbol{\theta}^*_{\mathrm{KL}}))dt \right] (\boldsymbol{\theta}^*_{\mathrm{KL}} - \hat{\boldsymbol{\theta}}_{\mathrm{KL}})$. Therefore, Equation (4) can be written as

$$\left[ \sum_{k=1}^{d} \int p(x|\hat{\boldsymbol{\theta}}_k) \int_0^1 \ddot{s}(\boldsymbol{x}; \boldsymbol{\theta}^*_{\mathrm{KL}} + t(\hat{\boldsymbol{\theta}}_{\mathrm{KL}} - \boldsymbol{\theta}^*_{\mathrm{KL}}))dt d\boldsymbol{x} \right] (\boldsymbol{\theta}^*_{\mathrm{KL}} - \hat{\boldsymbol{\theta}}_{\mathrm{KL}}) = o_p(\frac{1}{n}). \tag{5}$$

Under our Assumption 1, the Fish Information matrix $I(\boldsymbol{\theta})$ is positive definite in a neighborhood of $\boldsymbol{\theta}^*$, then we can find constant $C_1$, $C_2$ such that $C_1 \leq \| \int p(x|\hat{\boldsymbol{\theta}}_k) \int_0^1 \ddot{s}(\boldsymbol{x}; \boldsymbol{\theta}^*_{\mathrm{KL}} + t(\hat{\boldsymbol{\theta}}_{\mathrm{KL}} - \boldsymbol{\theta}^*_{\mathrm{KL}}))dt d\boldsymbol{x}\| \leq C_2$. Therefore, we can get $\mathbb{E}(\hat{\boldsymbol{\theta}}_{\mathrm{KL}} - \boldsymbol{\theta}^*_{\mathrm{KL}}) = o((dn)^{-1})$. $\square$

The following theorem provides the MSE between $\hat{\boldsymbol{\theta}}_{\mathrm{KL}}$ and $\boldsymbol{\theta}^*_{\mathrm{KL}}$ and that between $\hat{\boldsymbol{\theta}}_{\mathrm{KL}}$ and $\boldsymbol{\theta}^*$.

**Theorem 2.** *Based on Assumption 1, as $n \to \infty$, $\mathbb{E}\|\hat{\boldsymbol{\theta}}_{\mathrm{KL}} - \boldsymbol{\theta}^*_{\mathrm{KL}}\|^2 = O(\frac{1}{nd})$. Further, in terms of estimating the true parameter, we have*

$$\mathbb{E}\|\hat{\boldsymbol{\theta}}_{\mathrm{KL}} - \boldsymbol{\theta}^*\|^2 = O(N^{-1} + (dn)^{-1}). \tag{6}$$

**Proof:** According to the Equation (4),

$$\hat{\boldsymbol{\theta}}_{\mathrm{KL}} = \arg\max_{\boldsymbol{\theta} \in \Theta} \sum_{k=1}^{d} \frac{1}{n} \sum_{j=1}^{n} s(\widetilde{\boldsymbol{x}}_j^k; \boldsymbol{\theta}). \tag{7}$$

Then the first order derivative of Equation (7) with respect to $\boldsymbol{\theta}$ at $\boldsymbol{\theta} = \hat{\boldsymbol{\theta}}_{\mathrm{KL}}$ is zero,

$$\sum_{k=1}^{d} \frac{1}{n} \sum_{j=1}^{n} \dot{s}(\widetilde{\boldsymbol{x}}_j^k; \hat{\boldsymbol{\theta}}_{\mathrm{KL}}) = 0. \tag{8}$$

By Taylor expansion of Equation (8), we get

$$\sum_{k=1}^{d} \frac{1}{n} \sum_{j=1}^{n} (\dot{s}(\widetilde{\boldsymbol{x}}_j^k; \boldsymbol{\theta}^*_{\mathrm{KL}}) + \ddot{s}(\widetilde{\boldsymbol{x}}_j^k; \hat{\boldsymbol{\theta}}_{\mathrm{KL}})(\hat{\boldsymbol{\theta}}_{\mathrm{KL}} - \boldsymbol{\theta}^*_{\mathrm{KL}})) + o_p(\hat{\boldsymbol{\theta}}_{\mathrm{KL}} - \boldsymbol{\theta}^*_{\mathrm{KL}}) = 0.$$

By the law of large numbers, $\frac{1}{n}\sum_{j=1}^{n} \ddot{s}(\widetilde{\boldsymbol{x}}_j^k; \boldsymbol{\theta}^*_{\mathrm{KL}}) = I(\hat{\boldsymbol{\theta}}_k, \boldsymbol{\theta}^*_{\mathrm{KL}}) + o_p(\frac{1}{n})$. Under our Assumption 1, $I(\boldsymbol{\theta})$ is positive definite in a neighborhood of $\boldsymbol{\theta}^*$. Since $\hat{\boldsymbol{\theta}}_k$ are in the neighborhood of $\boldsymbol{\theta}^*$, $I(\hat{\boldsymbol{\theta}}_k, \boldsymbol{\theta}^*_{\mathrm{KL}})$ is positive definite, for $k = 1 \in [d]$. Then we have

$$\hat{\boldsymbol{\theta}}_{\mathrm{KL}} - \boldsymbol{\theta}^*_{\mathrm{KL}} = (\sum_{k=1}^{d} I(\hat{\boldsymbol{\theta}}_k, \boldsymbol{\theta}^*_{\mathrm{KL}}))^{-1} \sum_{k=1}^{d} \frac{1}{n} \sum_{j=1}^{n} \dot{s}(\widetilde{\boldsymbol{x}}_j^k; \boldsymbol{\theta}^*_{\mathrm{KL}}) + o_p(\frac{1}{n}) = 0. \tag{9}$$

By the central limit theorem, $\frac{1}{\sqrt{n}}\sum_{j=1}^{n} \dot{s}(\widetilde{\boldsymbol{x}}_j^k; \boldsymbol{\theta}^*_{\mathrm{KL}})$ converges to a normal distribution. By some simple calculation, we have

$$\mathrm{Cov}(\hat{\boldsymbol{\theta}}_{\mathrm{KL}} - \boldsymbol{\theta}^*_{\mathrm{KL}}, \hat{\boldsymbol{\theta}}_{\mathrm{KL}} - \boldsymbol{\theta}^*_{\mathrm{KL}}) = \frac{1}{n}(\sum_{k=1}^{d} I(\hat{\boldsymbol{\theta}}_k, \boldsymbol{\theta}^*_{\mathrm{KL}}))^{-1} \sum_{k=1}^{d} \mathrm{Var}(\dot{s}(\boldsymbol{x}; \boldsymbol{\theta}^*_{\mathrm{KL}}) \mid \hat{\boldsymbol{\theta}}_k)(\sum_{k=1}^{d} I(\hat{\boldsymbol{\theta}}_k, \boldsymbol{\theta}^*_{\mathrm{KL}}))^{-1}. \tag{10}$$

According to our Assumption 1, we already know $I(\hat{\boldsymbol{\theta}}_k, \boldsymbol{\theta}^*_{\mathrm{KL}})$ is positive definite, $C_1 \leq \|I(\hat{\boldsymbol{\theta}}_k, \boldsymbol{\theta}^*_{\mathrm{KL}})\| \leq C_2$. We have $(\sum_{k=1}^{d} I(\hat{\boldsymbol{\theta}}_k, \boldsymbol{\theta}^*_{\mathrm{KL}}))^{-1} = O(\frac{1}{d})$ and $\sum_{k=1}^{d} \mathrm{Var}(\dot{s}(\boldsymbol{x}; \boldsymbol{\theta}^*_{\mathrm{KL}}) \mid \hat{\boldsymbol{\theta}}_k) = O(d)$. Therefore, $\mathbb{E}\|\hat{\boldsymbol{\theta}}_{\mathrm{KL}} - \boldsymbol{\theta}^*_{\mathrm{KL}}\|^2 = \mathrm{trace}(\mathrm{Cov}(\hat{\boldsymbol{\theta}}_{\mathrm{KL}} - \boldsymbol{\theta}^*_{\mathrm{KL}}, \hat{\boldsymbol{\theta}}_{\mathrm{KL}} - \boldsymbol{\theta}^*_{\mathrm{KL}})) = O(\frac{1}{nd})$. Because the MSE between the exact KL estimator $\boldsymbol{\theta}^*_{\mathrm{KL}}$ and the true parameter $\boldsymbol{\theta}^*$ is $O(N^{-1})$ as shown in Liu and Ihler (2014), the MSE between $\hat{\boldsymbol{\theta}}_{\mathrm{KL}}$ and the true parameter $\boldsymbol{\theta}^*$ is

$$\mathbb{E}\|\hat{\boldsymbol{\theta}}_{\mathrm{KL}} - \boldsymbol{\theta}^*\|^2 \approx \mathbb{E}\|\hat{\boldsymbol{\theta}}_{\mathrm{KL}} - \boldsymbol{\theta}^*_{\mathrm{KL}}\|^2 + \mathbb{E}\|\boldsymbol{\theta}^*_{\mathrm{KL}} - \boldsymbol{\theta}^*\|^2 = O(N^{-1} + (dn)^{-1}).$$

We complete the proof of this theorem. $\square$

## 2  Appendix B

In this section, we analyze the MSE of our proposed estimator $\hat{\boldsymbol{\theta}}_{\mathrm{KL}-C}$ and prove Theorem 3.

**Theorem 3.** *Under Assumptions 1, we have*

$$as \ n \to \infty, \quad n\mathbb{E}\|\hat{\boldsymbol{\theta}}_{\mathrm{KL}-C} - \boldsymbol{\theta}^*_{\mathrm{KL}}\|^2 < n\mathbb{E}\|\hat{\boldsymbol{\theta}}_{\mathrm{KL}} - \boldsymbol{\theta}^*_{\mathrm{KL}}\|^2.$$

Since $\widetilde{\boldsymbol{\theta}}_k$ is the MLE of data $\{\widetilde{\boldsymbol{x}}_j^k\}_{j=1}^{n}$, then we have

$$(\widetilde{\boldsymbol{\theta}}_k - \hat{\boldsymbol{\theta}}_k) = -I(\hat{\boldsymbol{\theta}}_k)^{-1} \frac{1}{n} \sum_{j=1}^{n} \dot{s}(\widetilde{\boldsymbol{x}}_j^k; \hat{\boldsymbol{\theta}}_k) + o_p(\frac{1}{n}). \tag{11}$$

Then $\mathbb{E}(\widetilde{\boldsymbol{\theta}}_k - \hat{\boldsymbol{\theta}}_k) = o(\frac{1}{n})$. According to Theorem (2), when $\mathfrak{B}_k$ is a constant matrix, for $k \in [d]$,

$$\mathbb{E}(\hat{\boldsymbol{\theta}}_{\text{KL}-C} - \boldsymbol{\theta}^*_{\text{KL}}) = \mathbb{E}(\hat{\boldsymbol{\theta}}_{\text{KL}} - \boldsymbol{\theta}^*_{\text{KL}}) + \sum_{k=1}^{d} \mathfrak{B}_k \mathbb{E}(\widetilde{\boldsymbol{\theta}}_k - \hat{\boldsymbol{\theta}}_k) = o(\frac{1}{n}).$$

Notice that $\frac{1}{n}\sum_{j=1}^{n} \dot{\text{s}}(\widetilde{\boldsymbol{x}}_j^r; \hat{\boldsymbol{\theta}}_r)$ and $\frac{1}{n}\sum_{j=1}^{n} \dot{\text{s}}(\widetilde{\boldsymbol{x}}_j^t; \hat{\boldsymbol{\theta}}_t)$ are independent when $r \neq t$. According to Equation (9), we know $\sum_{k=1}^{d} \frac{1}{n}\sum_{j=1}^{n} \dot{\text{s}}(\widetilde{\boldsymbol{x}}_j^k; \boldsymbol{\theta}^*_{\text{KL}})$ and $\frac{1}{n}\sum_{j=1}^{n} \dot{\text{s}}(\widetilde{\boldsymbol{x}}_j^k; \hat{\boldsymbol{\theta}}_k)$ are correlated to each other for $k \in [d]$,

$$\text{Cov}((\hat{\boldsymbol{\theta}}_{\text{KL}-C} - \boldsymbol{\theta}^*_{\text{KL}}), (\hat{\boldsymbol{\theta}}_{\text{KL}-C} - \boldsymbol{\theta}^*_{\text{KL}})) = \text{Cov}(\hat{\boldsymbol{\theta}}_{\text{KL}} - \boldsymbol{\theta}^*_{\text{KL}}, \hat{\boldsymbol{\theta}}_{\text{KL}} - \boldsymbol{\theta}^*_{\text{KL}})$$
$$+ 2\sum_{k=1}^{d} \mathfrak{B}_k \text{Cov}(\hat{\boldsymbol{\theta}}_{\text{KL}} - \boldsymbol{\theta}_{\text{KL}}, \widetilde{\boldsymbol{\theta}}_k - \hat{\boldsymbol{\theta}}_k)^T + \sum_{k=1}^{d} \mathfrak{B}_k \text{Cov}((\widetilde{\boldsymbol{\theta}}_k - \hat{\boldsymbol{\theta}}_k), (\widetilde{\boldsymbol{\theta}}_k - \hat{\boldsymbol{\theta}}_k)) \mathfrak{B}_k^T.$$

When $\boldsymbol{B}_k = -(\text{Cov}(\widetilde{\boldsymbol{\theta}}_k - \hat{\boldsymbol{\theta}}_k, \widetilde{\boldsymbol{\theta}}_k - \hat{\boldsymbol{\theta}}_k))^{-1}\text{Cov}(\hat{\boldsymbol{\theta}}_{\text{KL}} - \boldsymbol{\theta}^*_{\text{KL}}, \widetilde{\boldsymbol{\theta}}_k - \hat{\boldsymbol{\theta}}_k)$, we have

$$\text{Cov}(\hat{\boldsymbol{\theta}}_{\text{KL}-C} - \boldsymbol{\theta}^*_{KL}, \hat{\boldsymbol{\theta}}_{\text{KL}-C} - \boldsymbol{\theta}^*_{\text{KL}}) = \text{Cov}(\hat{\boldsymbol{\theta}}_{\text{KL}} - \boldsymbol{\theta}^*_{\text{KL}}, \hat{\boldsymbol{\theta}}_{\text{KL}} - \boldsymbol{\theta}^*_{\text{KL}}) -$$
$$\sum_{k=1}^{d} \text{Cov}(\widetilde{\boldsymbol{\theta}}_k - \hat{\boldsymbol{\theta}}_k, \widetilde{\boldsymbol{\theta}}_k - \hat{\boldsymbol{\theta}}_k)^{-1}\text{Cov}(\hat{\boldsymbol{\theta}}_{KL} - \boldsymbol{\theta}^*_{\text{KL}}, \widetilde{\boldsymbol{\theta}}_k - \hat{\boldsymbol{\theta}}_k)\text{Cov}(\hat{\boldsymbol{\theta}}_{\text{KL}} - \boldsymbol{\theta}^*_{\text{KL}}, \widetilde{\boldsymbol{\theta}}_k - \hat{\boldsymbol{\theta}}_k)^T. \quad (12)$$

We know $\mathbb{E}\|\hat{\boldsymbol{\theta}}_{\text{KL}-C} - \boldsymbol{\theta}^*_{\text{KL}}\|^2 = \text{trace}(\text{Cov}(\hat{\boldsymbol{\theta}}_{\text{KL}-C} - \boldsymbol{\theta}^*_{\text{KL}}, \hat{\boldsymbol{\theta}}_{\text{KL}-C} - \boldsymbol{\theta}^*_{\text{KL}})), \mathbb{E}\|\hat{\boldsymbol{\theta}}_{\text{KL}} - \boldsymbol{\theta}^*_{\text{KL}}\|^2 = \text{trace}(\text{Cov}(\hat{\boldsymbol{\theta}}_{\text{KL}} - \boldsymbol{\theta}^*_{\text{KL}}, \hat{\boldsymbol{\theta}}_{\text{KL}} - \boldsymbol{\theta}^*_{\text{KL}}))$. The second term of Equation (12) is a positive definite matrix, therefore we have $n\mathbb{E}\|\hat{\boldsymbol{\theta}}_{\text{KL}-C} - \boldsymbol{\theta}^*_{\text{KL}}\|^2 < n\mathbb{E}\|\hat{\boldsymbol{\theta}}_{\text{KL}} - \boldsymbol{\theta}^*_{\text{KL}}\|^2$ as $n \to \infty$. We complete the proof of this theorem. $\square$

**Theorem 4.** *Under Assumption 1, when $N > n \times d$, we have $E\|\hat{\boldsymbol{\theta}}_{\text{KL}-C} - \boldsymbol{\theta}^*_{\text{KL}}\|^2 = O(\frac{1}{dn^2})$ as $n \to \infty$. Further, in terms of estimating the true parameter, we have*

$$\mathbb{E}\|\hat{\boldsymbol{\theta}}_{\text{KL}-C} - \boldsymbol{\theta}^*\|^2 = O(N^{-1} + (dn^2)^{-1}).$$

From Equation (4), we know

$$\sum_{k=1}^{d} \frac{1}{n} \sum_{j=1}^{n} \frac{\partial \log p(\widetilde{\boldsymbol{x}}_j^k | \hat{\boldsymbol{\theta}}_{\text{KL}})}{\partial \boldsymbol{\theta}} = 0. \quad (13)$$

By Taylor expansion, Equation (13) can be rewritten as

$$\sum_{k=1}^{d} \left[\frac{1}{n} \sum_{j=1}^{n} \dot{\text{s}}(\widetilde{\boldsymbol{x}}_j^k; \hat{\boldsymbol{\theta}}_k) + \ddot{\text{s}}(\widetilde{\boldsymbol{x}}_j^k; \hat{\boldsymbol{\theta}}_k)(\hat{\boldsymbol{\theta}}_{\text{KL}} - \hat{\boldsymbol{\theta}}_k)) + O_p(\|\hat{\boldsymbol{\theta}}_{\text{KL}} - \hat{\boldsymbol{\theta}}_k\|^2)\right] = 0. \quad (14)$$

$\|\hat{\boldsymbol{\theta}}_{\text{KL}} - \hat{\boldsymbol{\theta}}_k\|^2 \leq \|\hat{\boldsymbol{\theta}}_{\text{KL}} - \boldsymbol{\theta}^*_{\text{KL}}\|^2 + \|\boldsymbol{\theta}^*_{\text{KL}} - \hat{\boldsymbol{\theta}}_k\|^2$. As we know from Liu and Ihler (2014), we have

$$\|\boldsymbol{\theta}^*_{\text{KL}} - \hat{\boldsymbol{\theta}}_k\|^2 \leq \|\boldsymbol{\theta}^*_{\text{KL}} - \boldsymbol{\theta}^*\|^2 + \|\boldsymbol{\theta}^* - \hat{\boldsymbol{\theta}}_k\|^2 = O_p(\frac{d}{N}), \quad (15)$$

When $N > n \times d$, we have $\|\hat{\boldsymbol{\theta}}_{\text{KL}} - \hat{\boldsymbol{\theta}}_k\|^2 = O_p(\frac{1}{nd})$. And it is also easy to derive

$$\hat{\boldsymbol{\theta}}_{\text{KL}} - \hat{\boldsymbol{\theta}}_k = \hat{\boldsymbol{\theta}}_{\text{KL}} - \boldsymbol{\theta}^*_{\text{KL}} + \boldsymbol{\theta}^*_{\text{KL}} - \boldsymbol{\theta}^* + \boldsymbol{\theta}^* - \hat{\boldsymbol{\theta}}_k = o_p(\frac{1}{N}) + o_p(\frac{1}{N}) + o_p(\frac{d}{N}) = o_p(\frac{1}{nd} + \frac{d}{N}), \quad (16)$$

where $\boldsymbol{\theta}^*_{KL} - \boldsymbol{\theta}^* = o_p(\frac{1}{N})$ has been proved in Liu and Ihler's paper(2014). According to the law of large numbers, $\frac{1}{n}\sum_{j=1}^{n} \ddot{\text{s}}(\widetilde{\boldsymbol{x}}_j^k; \hat{\boldsymbol{\theta}}_k) = I(\hat{\boldsymbol{\theta}}_k) + o_p(\frac{1}{n})$, then we have

$$(\hat{\boldsymbol{\theta}}_{\text{KL}} - \boldsymbol{\theta}^*_{\text{KL}}) = -(\sum_{k=1}^{d} I(\hat{\boldsymbol{\theta}}_k))^{-1} \sum_{k=1}^{d} \frac{1}{n} \sum_{j=1}^{n} \dot{\text{s}}(\widetilde{\boldsymbol{x}}_j^k; \hat{\boldsymbol{\theta}}_k) + O_p(\frac{1}{nd}). \quad (17)$$

Notie that $\frac{1}{n}\sum_{j=1}^{n}\dot{s}(\widetilde{\boldsymbol{x}}_{j}^{r};\hat{\boldsymbol{\theta}}_{r})$ and $\frac{1}{n}\sum_{j=1}^{n}\dot{s}(\widetilde{\boldsymbol{x}}_{j}^{t};\hat{\boldsymbol{\theta}}_{t})$ are independent when $r \neq t$. Therefore from (11) and (17), the covariance matrix of $n(\hat{\boldsymbol{\theta}}_{\text{KL}} - \boldsymbol{\theta}_{\text{KL}}^{*})$ and $n(\widetilde{\boldsymbol{\theta}}_{k} - \hat{\boldsymbol{\theta}}_{k})$ is

$$\text{Cov}(n(\hat{\boldsymbol{\theta}}_{\text{KL}} - \boldsymbol{\theta}_{\text{KL}}^{*}), n(\widetilde{\boldsymbol{\theta}}_{k} - \hat{\boldsymbol{\theta}}_{k})) = n(\sum_{k=1}^{d} I(\hat{\boldsymbol{\theta}}_{k}))^{-1} + (\sum_{k=1}^{d} I(\hat{\boldsymbol{\theta}}_{k}))^{-1}O(1),$$

for $k \in [d]$. According to Assumption 1, we know $\sum_{k=1}^{d} I(\hat{\boldsymbol{\theta}}_{k}) = O(d)$. Then we will have

$$\text{Cov}(n(\hat{\boldsymbol{\theta}}_{\text{KL}} - \boldsymbol{\theta}_{\text{KL}}^{*}), n(\widetilde{\boldsymbol{\theta}}_{k} - \hat{\boldsymbol{\theta}}_{k})) = n(\sum_{k=1}^{d} I(\hat{\boldsymbol{\theta}}_{k}))^{-1} + O(\frac{1}{d}), \quad \text{for } k \in [d]. \tag{18}$$

According to Theorem 2 and Equation (10), by the law of large numbers, it is easy to derive

$$\text{Cov}(n(\hat{\boldsymbol{\theta}}_{\text{KL}} - \boldsymbol{\theta}_{KL}^{*}), n(\hat{\boldsymbol{\theta}}_{\text{KL}} - \boldsymbol{\theta}_{\text{KL}}^{*})) = n(\sum_{k=1}^{d} I(\hat{\boldsymbol{\theta}}_{k}))^{-1} + o(1).$$

$$\text{Cov}(n(\hat{\boldsymbol{\theta}}_{\text{KL}-C} - \boldsymbol{\theta}_{\text{KL}}^{*}), n(\hat{\boldsymbol{\theta}}_{\text{KL}-C} - \boldsymbol{\theta}_{\text{KL}}^{*})) = \text{Cov}(n(\hat{\boldsymbol{\theta}}_{\text{KL}} - \boldsymbol{\theta}_{\text{KL}}^{*}), n(\hat{\boldsymbol{\theta}}_{\text{KL}} - \boldsymbol{\theta}_{\text{KL}}^{*})$$
$$+ 2\sum_{k=1}^{d} \mathfrak{B}_{k}\text{Cov}(n(\hat{\boldsymbol{\theta}}_{\text{KL}} - \boldsymbol{\theta}_{\text{KL}}^{*}), n(\widetilde{\boldsymbol{\theta}}_{k} - \hat{\boldsymbol{\theta}}_{k}))^{\top} + \sum_{k=1}^{d} \mathfrak{B}_{k}\text{Cov}(n(\widetilde{\boldsymbol{\theta}}_{k} - \hat{\boldsymbol{\theta}}_{k}), n(\widetilde{\boldsymbol{\theta}}_{k} - \hat{\boldsymbol{\theta}}_{k}))\mathfrak{B}_{k}^{T},$$
$$\tag{19}$$

where $\mathfrak{B}_{k}$ is defined in (8),

$$\mathfrak{B}_{k} = -(\sum_{k=1}^{d} I(\hat{\boldsymbol{\theta}}_{k}))^{-1} I(\hat{\boldsymbol{\theta}}_{k}), \quad k \in [d].$$

According to Equation (11), we know $\text{Cov}(n(\widetilde{\boldsymbol{\theta}}_{k} - \hat{\boldsymbol{\theta}}_{k}), n(\widetilde{\boldsymbol{\theta}}_{k} - \hat{\boldsymbol{\theta}}_{k})) = n(I(\hat{\boldsymbol{\theta}}_{k}))^{-1} + o(1)$. By some simple calculation, we know that $n^{2}\text{Cov}(\hat{\boldsymbol{\theta}}_{\text{KL}-C} - \boldsymbol{\theta}_{\text{KL}}^{*}, \hat{\boldsymbol{\theta}}_{\text{KL}-C} - \boldsymbol{\theta}_{\text{KL}}^{*}) = O(\frac{1}{d})$. Therefore, under the Assumption 1, when $N > n \times d$, we get the following result,

$$\mathbb{E}\|\hat{\boldsymbol{\theta}}_{\text{KL}-C} - \boldsymbol{\theta}_{\text{KL}}^{*}\|^{2} = \text{trace}(\text{Cov}(\hat{\boldsymbol{\theta}}_{\text{KL}-C} - \boldsymbol{\theta}_{\text{KL}}^{*}, \hat{\boldsymbol{\theta}}_{\text{KL}-C} - \boldsymbol{\theta}_{\text{KL}}^{*})) = O(\frac{1}{dn^{2}}).$$

We know $\mathbb{E}\|\boldsymbol{\theta}_{\text{KL}}^{*} - \boldsymbol{\theta}^{*}\|^{2} = O(N^{-1})$ from Liu and Ihler (2014). Then we have

$$\mathbb{E}\|\hat{\boldsymbol{\theta}}_{\text{KL}-C} - \boldsymbol{\theta}^{*}\|^{2} \approx \mathbb{E}\|\hat{\boldsymbol{\theta}}_{\text{KL}-C} - \boldsymbol{\theta}_{\text{KL}}^{*}\|^{2} + \mathbb{E}\|\boldsymbol{\theta}_{\text{KL}}^{*} - \boldsymbol{\theta}^{*}\|^{2} = O(N^{-1} + (dn^{2})^{-1}).$$

The proof of this theorem is complete. □

## 3  Appendix C

In this section, we analyze the asymptotic property of $\hat{\boldsymbol{\theta}}_{\text{KL}-W}$ and prove Theorem 5. We show the MSE between $\hat{\boldsymbol{\theta}}_{\text{KL}-W}$ and $\boldsymbol{\theta}_{\text{KL}}^{*}$ is much smaller than the MSE between the KL-naive estimator $\hat{\boldsymbol{\theta}}_{\text{KL}}$ and $\boldsymbol{\theta}_{\text{KL}}^{*}$.

**Lemma 5.** *Under Assumption 1, as $n \to \infty$, $\widetilde{\eta}(\boldsymbol{\theta})$ is a more accurate estimator of $\eta(\boldsymbol{\theta})$ than $\hat{\eta}(\boldsymbol{\theta})$, i.e.,*

$$n\text{Var}(\widetilde{\eta}(\boldsymbol{\theta})) \leq n\text{Var}(\hat{\eta}(\boldsymbol{\theta})), \quad \text{for any } \boldsymbol{\theta} \in \Theta. \tag{20}$$

By Taylor expansion,

$$\frac{p(\boldsymbol{x}|\hat{\boldsymbol{\theta}}_{k})}{p(\boldsymbol{x}|\widetilde{\boldsymbol{\theta}}_{k})} = 1 + (\log p(\boldsymbol{x}|\hat{\boldsymbol{\theta}}_{k}) - \log p(\boldsymbol{x}|\widetilde{\boldsymbol{\theta}}_{k})) + O_{p}(\|\widetilde{\boldsymbol{\theta}}_{k} - \hat{\boldsymbol{\theta}}_{k}\|^{2}), \tag{21}$$

we will have

$$\widetilde{\eta}(\boldsymbol{\theta}) = \sum_{k=1}^{d}[\frac{1}{n}\sum_{j=1}^{n}(1 + (s(\widetilde{\boldsymbol{x}}_{j}^{k};\hat{\boldsymbol{\theta}}_{k}) - s(\widetilde{\boldsymbol{x}}_{j}^{k};\widetilde{\boldsymbol{\theta}}_{k})))s(\widetilde{\boldsymbol{x}}_{j}^{k};\boldsymbol{\theta}) + O_{p}(\|\widetilde{\boldsymbol{\theta}}_{k} - \hat{\boldsymbol{\theta}}_{k}\|^{2})],$$

Since $s(\boldsymbol{x}; \hat{\boldsymbol{\theta}}_k) - s(\boldsymbol{x}; \widetilde{\boldsymbol{\theta}}_k) = \dot{s}(\boldsymbol{x}; \hat{\boldsymbol{\theta}}_k)(\hat{\boldsymbol{\theta}}_k - \widetilde{\boldsymbol{\theta}}_k)$, according to equation (11), we have

$$\widetilde{\eta}(\boldsymbol{\theta}) = \hat{\eta}(\boldsymbol{\theta}) - \sum_{k=1}^{d} \frac{1}{n} \sum_{j=1}^{n} s(\widetilde{\boldsymbol{x}}_j^k; \boldsymbol{\theta}) \dot{s}(\widetilde{\boldsymbol{x}}_j^k; \hat{\boldsymbol{\theta}}_k)(\widetilde{\boldsymbol{\theta}}_k - \hat{\boldsymbol{\theta}}^k) + O_p(\|\widetilde{\boldsymbol{\theta}}_k - \hat{\boldsymbol{\theta}}_k\|^2),$$

Then according to equation (11), we have

$$\hat{\eta}(\boldsymbol{\theta}) = \widetilde{\eta}(\boldsymbol{\theta}) - \sum_{k=1}^{d} \mathbb{E}(s(\widetilde{\boldsymbol{x}}_j^k; \boldsymbol{\theta}) \dot{s}(\widetilde{\boldsymbol{x}}_j^k; \hat{\boldsymbol{\theta}}_k) \mid \hat{\boldsymbol{\theta}}_k)) I(\hat{\boldsymbol{\theta}}_k)^{-1} \frac{1}{n} \sum_{j=1}^{n} \dot{s}(\widetilde{\boldsymbol{x}}_j^k; \hat{\boldsymbol{\theta}}_k) + O_p(\frac{d}{n}),$$

Denote $\hat{\xi}(\boldsymbol{\theta}) = -\sum_{k=1}^{d} \mathbb{E}(s(\widetilde{\boldsymbol{x}}_j^k; \boldsymbol{\theta}) \dot{s}(\widetilde{\boldsymbol{x}}_j^k; \hat{\boldsymbol{\theta}}_k) \mid \hat{\boldsymbol{\theta}}_k)) I(\hat{\boldsymbol{\theta}}_k)^{-1} \frac{1}{n} \sum_{j=1}^{n} \dot{s}(\boldsymbol{x}_j^k; \hat{\boldsymbol{\theta}}_k)$. According to Henmi et al. (2007), $\hat{\xi}(\boldsymbol{\theta})$ is the orthogonal projection of $\hat{\eta}(\boldsymbol{\theta})$ onto the linear space spanned by the score vector component for each $\hat{\boldsymbol{\theta}}_k$, where $k \in [d]$. Then we will have $\mathrm{Var}(\hat{\eta}(\boldsymbol{\theta})) = \mathrm{Var}(\widetilde{\eta}(\boldsymbol{\theta})) + \mathrm{Var}(\hat{\xi}(\boldsymbol{\theta}))$. Therefore, $n\mathrm{Var}(\widetilde{\eta}(\boldsymbol{\theta})) \leq n\mathrm{Var}(\hat{\eta}(\boldsymbol{\theta}))$.

**Theorem 6.** *Under the Assumption 1, for any $\{\hat{\boldsymbol{\theta}}_k\}$, we have that*

$$\text{as } n \to \infty, \quad n\mathbb{E}\|\hat{\boldsymbol{\theta}}_{\mathrm{KL-W}} - \boldsymbol{\theta}_{\mathrm{KL}}^*\|^2 \leq n\mathbb{E}\|\hat{\boldsymbol{\theta}}_{\mathrm{KL}} - \boldsymbol{\theta}_{\mathrm{KL}}^*\|^2.$$

**Proof:** From Equation (10), we know

$$\sum_{k=1}^{d} \frac{1}{n} \sum_{j=1}^{n} \frac{p(\widetilde{\boldsymbol{x}}_j^k|\hat{\boldsymbol{\theta}}_k)}{p(\widetilde{\boldsymbol{x}}_j^k|\widetilde{\boldsymbol{\theta}}_k)} \dot{s}(\widetilde{\boldsymbol{x}}_j^k; \hat{\boldsymbol{\theta}}_{\mathrm{KL-W}}) = 0.$$

Since $\frac{p(\boldsymbol{x}|\hat{\boldsymbol{\theta}}_k)}{p(\boldsymbol{x}|\widetilde{\boldsymbol{\theta}}_k)} = \exp\{\log p(\boldsymbol{x}|\hat{\boldsymbol{\theta}}_k) - \log p(\boldsymbol{x}|\widetilde{\boldsymbol{\theta}}_k)\} = 1 + (\log p(\boldsymbol{x}|\hat{\boldsymbol{\theta}}_k) - \log p(\boldsymbol{x}|\widetilde{\boldsymbol{\theta}}_k)) + O_p(\|\widetilde{\boldsymbol{\theta}}_k - \hat{\boldsymbol{\theta}}_k\|^2)$, we have

$$\sum_{k=1}^{d} \frac{1}{n} \sum_{j=1}^{n} \dot{s}(\boldsymbol{x}_j^k; \hat{\boldsymbol{\theta}}_{\mathrm{KL-W}}) - \sum_{k=1}^{d} [\frac{1}{n} \sum_{j=1}^{n} \dot{s}(\boldsymbol{x}_j^k; \hat{\boldsymbol{\theta}}_{\mathrm{KL-W}}) \dot{s}(\boldsymbol{x}_j^k; \hat{\boldsymbol{\theta}}_k)^T (\widetilde{\boldsymbol{\theta}}_k - \hat{\boldsymbol{\theta}}_k) + O_p(\|\widetilde{\boldsymbol{\theta}}_k - \hat{\boldsymbol{\theta}}_k\|^2)] = 0.$$
(22)

From the asymptotic property of MLE, we know $\mathbb{E}\|\widetilde{\boldsymbol{\theta}}_k - \hat{\boldsymbol{\theta}}_k\|^2 = \frac{1}{n}\mathrm{trace}(I(\hat{\boldsymbol{\theta}}_k))$. Therefore, we know $\|\widetilde{\boldsymbol{\theta}}_k - \hat{\boldsymbol{\theta}}_k\|^2 = O_p(\frac{1}{n})$ and $\sum_{k=1}^{d} \|\widetilde{\boldsymbol{\theta}}_k - \hat{\boldsymbol{\theta}}_k\|^2 = O_p(\frac{d}{n})$.

Similar to the derivation of equation (9), according to equation (11), we have the following equation,

$$\hat{\boldsymbol{\theta}}_{\mathrm{KL-W}} - \boldsymbol{\theta}_{\mathrm{KL}}^* = (\sum_{k=1}^{d} I(\hat{\boldsymbol{\theta}}_k, \boldsymbol{\theta}_{\mathrm{KL}}^*))^{-1} \sum_{k=1}^{d} \frac{1}{n} \sum_{j=1}^{n} \dot{s}(\widetilde{\boldsymbol{x}}_j^k; \boldsymbol{\theta}_{\mathrm{KL}}^*) -$$

$$(\sum_{k=1}^{d} I(\hat{\boldsymbol{\theta}}_k, \boldsymbol{\theta}_{\mathrm{KL}}^*))^{-1} \sum_{k=1}^{d} \mathbb{E}(\dot{s}(\widetilde{\boldsymbol{x}}_j^k; \hat{\boldsymbol{\theta}}_{\mathrm{KL-W}})^T \dot{s}(\widetilde{\boldsymbol{x}}_j^k; \hat{\boldsymbol{\theta}}_k) \mid \hat{\boldsymbol{\theta}}_k) \frac{1}{n} \sum_{j=1}^{n} \dot{s}(\widetilde{\boldsymbol{x}}_j^k; \hat{\boldsymbol{\theta}}_k) = O_p(\frac{d}{n}).$$

Then we have,

$$\hat{\boldsymbol{\theta}}_{\mathrm{KL}} - \boldsymbol{\theta}_{\mathrm{KL}}^* = \hat{\boldsymbol{\theta}}_{\mathrm{KL-W}} - \boldsymbol{\theta}_{\mathrm{KL}}^*$$

$$- (\sum_{k=1}^{d} I(\hat{\boldsymbol{\theta}}_k, \boldsymbol{\theta}_{\mathrm{KL}}^*))^{-1} \sum_{k=1}^{d} \mathbb{E}(\dot{s}(\widetilde{\boldsymbol{x}}_j^k; \hat{\boldsymbol{\theta}}_{\mathrm{KL-W}})^T \dot{s}(\widetilde{\boldsymbol{x}}_j^k; \hat{\boldsymbol{\theta}}_k) \mid \hat{\boldsymbol{\theta}}_k) \frac{1}{n} \sum_{j=1}^{n} \dot{s}(\widetilde{\boldsymbol{x}}_j^k; \hat{\boldsymbol{\theta}}_k) = O_p(\frac{d}{n}).$$

According to Henmi et al.(2007), we know the second term of above equation is the orthogonal projection of $(\hat{\boldsymbol{\theta}}_{\mathrm{KL}} - \boldsymbol{\theta}_{\mathrm{KL}}^*)$ onto the linear space spanned by the score component for each $\hat{\boldsymbol{\theta}}_k$, for $k \in [d]$. Then

$$n\mathbb{E}\|\hat{\boldsymbol{\theta}}_{\mathrm{KL-W}} - \boldsymbol{\theta}_{\mathrm{KL}}^*\|^2 \leq n\mathbb{E}\|\hat{\boldsymbol{\theta}}_{\mathrm{KL}} - \boldsymbol{\theta}_{KL}^*\|^2.$$

We complete the proof of this theorem. $\square$

**Theorem 7.** *Under the Assumptions 1, when $N > n \times d$, $\mathbb{E}\|\hat{\boldsymbol{\theta}}_{\mathrm{KL-W}} - \boldsymbol{\theta}_{\mathrm{KL}}^*\|^2 = O(\frac{1}{dn^2})$. Further, its MSE for estimating the true parameter $\boldsymbol{\theta}^*$ is*

$$\mathbb{E}\|\hat{\boldsymbol{\theta}}_{\mathrm{KL-W}} - \boldsymbol{\theta}^*\|^2 = O(N^{-1} + (dn^2)^{-1}).$$

According to Equation (22),

$$\sum_{k=1}^{d} \frac{1}{n} \sum_{j=1}^{n} \dot{s}(\widetilde{\boldsymbol{x}}_j^k; \hat{\boldsymbol{\theta}}_{\mathrm{KL}-W}) - \sum_{k=1}^{d} \frac{1}{n} \sum_{j=1}^{n} \dot{s}(\widetilde{\boldsymbol{x}}_j^k; \hat{\boldsymbol{\theta}}_{\mathrm{KL}-W}) \dot{s}(\widetilde{\boldsymbol{x}}_j^k; \hat{\boldsymbol{\theta}}_k)^T (\widetilde{\boldsymbol{\theta}}_k - \hat{\boldsymbol{\theta}}_k) = O_p(\frac{d}{n}).$$

Approximating the first term of the above equation by Taylor expansion, we will get

$$\sum_{k=1}^{d} \frac{1}{n} \sum_{j=1}^{n} \dot{s}(\widetilde{\boldsymbol{x}}_j^k; \hat{\boldsymbol{\theta}}_{\mathrm{KL}-W}) = \sum_{k=1}^{d} [\frac{1}{n} \sum_{j=1}^{n} \dot{s}(\widetilde{\boldsymbol{x}}_j^k; \hat{\boldsymbol{\theta}}_k)$$
$$+ \sum_{k=1}^{d} \frac{1}{n} \sum_{j=1}^{n} \ddot{s}(\widetilde{\boldsymbol{x}}_j^k; \hat{\boldsymbol{\theta}}_k)(\hat{\boldsymbol{\theta}}_{\mathrm{KL}-W} - \hat{\boldsymbol{\theta}}_k) + O_p(\|\hat{\boldsymbol{\theta}}_{\mathrm{KL}-W} - \hat{\boldsymbol{\theta}}_k\|^2)]. \tag{23}$$

Since $\|\hat{\boldsymbol{\theta}}_{\mathrm{KL}-W} - \hat{\boldsymbol{\theta}}_k\|^2 \le \|\hat{\boldsymbol{\theta}}_{\mathrm{KL}-W} - \boldsymbol{\theta}_{\mathrm{KL}}^*\|^2 + \|\boldsymbol{\theta}_{\mathrm{KL}}^* - \hat{\boldsymbol{\theta}}_k\|^2$, according to equation (15), then $\|\hat{\boldsymbol{\theta}}_{\mathrm{KL}-W} - \hat{\boldsymbol{\theta}}_k\|^2 = O_p(\|\hat{\boldsymbol{\theta}}_{\mathrm{KL}-W} - \boldsymbol{\theta}_{\mathrm{KL}}^*\|^2 + \frac{d}{N})$. We can easily derive $\dot{s}(\widetilde{\boldsymbol{x}}_j^k; \hat{\boldsymbol{\theta}}_{\mathrm{KL}-W}) = \dot{s}(\widetilde{\boldsymbol{x}}_j^k; \hat{\boldsymbol{\theta}}_k) + O_p(\hat{\boldsymbol{\theta}}_{\mathrm{KL}-W} - \hat{\boldsymbol{\theta}}_k)$ for $k \in [d]$. When $N > n \times d$, we will have

$$\sum_{k=1}^{d} \frac{1}{n} \sum_{j=1}^{n} \dot{s}(\widetilde{\boldsymbol{x}}_j^k; \hat{\boldsymbol{\theta}}_k) + \sum_{k=1}^{d} \frac{1}{n} \sum_{j=1}^{n} \ddot{s}(\widetilde{\boldsymbol{x}}_j^k; \hat{\boldsymbol{\theta}}_k)(\hat{\boldsymbol{\theta}}_{\mathrm{KL}-W} - \hat{\boldsymbol{\theta}}_k)$$
$$- \sum_{k} \frac{1}{n} \sum_{j=1}^{n} \dot{s}(\boldsymbol{x}_j^k; \hat{\boldsymbol{\theta}}_k) \dot{s}(\widetilde{\boldsymbol{x}}_j^k; \hat{\boldsymbol{\theta}}_k)^T (\widetilde{\boldsymbol{\theta}}_k - \hat{\boldsymbol{\theta}}_k) + O_p(\|\hat{\boldsymbol{\theta}}_{\mathrm{KL}-W} - \boldsymbol{\theta}_{\mathrm{KL}}^*\|^2) = O(\frac{d}{n}). \tag{24}$$

$\frac{1}{n} \sum_{j=1}^{n} \ddot{s}(\widetilde{\boldsymbol{x}}_j^k; \hat{\boldsymbol{\theta}}_k) = I(\hat{\boldsymbol{\theta}}_k) + o_p(\frac{1}{n})$ and we also know that $\frac{1}{n} \sum_{j=1}^{n} \dot{s}(\widetilde{\boldsymbol{x}}_j^k; \hat{\boldsymbol{\theta}}_k) \dot{s}(\widetilde{\boldsymbol{x}}_j^k; \hat{\boldsymbol{\theta}}_k)^T = I(\hat{\boldsymbol{\theta}}_k) + o_p(1)$. From (16), we know $\boldsymbol{\theta}_{\mathrm{KL}}^* - \hat{\boldsymbol{\theta}}_k = o_p(\frac{d}{N}) = o_p(\frac{1}{n})$. When $N > n \times d$, we have

$$\sum_{k=1}^{d} \frac{1}{n} \sum_{j=1}^{n} \dot{s}(\widetilde{\boldsymbol{x}}_j^k; \hat{\boldsymbol{\theta}}_k) + \sum_{k=1}^{d} I(\hat{\boldsymbol{\theta}}_k)(\hat{\boldsymbol{\theta}}_{\mathrm{KL}-W} - \boldsymbol{\theta}_{\mathrm{KL}}^*)$$
$$+ \sum_{k=1}^{d} \frac{1}{n} I(\hat{\boldsymbol{\theta}}_k)(\widetilde{\boldsymbol{\theta}}_k - \hat{\boldsymbol{\theta}}_k)) + O_p(\|\hat{\boldsymbol{\theta}}_{\mathrm{KL}-W} - \boldsymbol{\theta}_{\mathrm{KL}}^*\|^2) = O(\frac{d}{n}). \tag{25}$$

Based on the Equation (11), the first term and the third term of Equation (25) are cancelled. By some simple calculation, we will get

$$n^2 (\hat{\boldsymbol{\theta}}_{\mathrm{KL}-W} - \boldsymbol{\theta}_{\mathrm{KL}}^*)^T (\sum_{k=1}^{d} I(\hat{\boldsymbol{\theta}}_k))(\sum_{k=1}^{d} I(\hat{\boldsymbol{\theta}}_k))(\hat{\boldsymbol{\theta}}_{\mathrm{KL}-W} - \boldsymbol{\theta}_{\mathrm{KL}}^*) = O_p(d). \tag{26}$$

This indicates, $\mathrm{Cov}(n(\sum_{k=1}^{d} I(\hat{\boldsymbol{\theta}}_k))(\hat{\boldsymbol{\theta}}_{\mathrm{KL}-W} - \boldsymbol{\theta}_{\mathrm{KL}}^*), n(\sum_{k=1}^{d} I(\hat{\boldsymbol{\theta}}_k))(\hat{\boldsymbol{\theta}}_{\mathrm{KL}-W} - \boldsymbol{\theta}_{\mathrm{KL}}^*)) = O(d)$ as $n \to \infty$. We know $n^2 \mathbb{E}\|\hat{\boldsymbol{\theta}}_{\mathrm{KL}-W} - \boldsymbol{\theta}_{\mathrm{KL}}^*\|^2 = \mathrm{trace}(\mathrm{Cov}(n(\hat{\boldsymbol{\theta}}_{\mathrm{KL}-W} - \boldsymbol{\theta}_{\mathrm{KL}}^*), n(\hat{\boldsymbol{\theta}}_{\mathrm{KL}-W} - \boldsymbol{\theta}_{\mathrm{KL}}^*)))$. According to Assumption 1, $I(\hat{\boldsymbol{\theta}}_k)$ is positive definite and then $\mathrm{trace}(\sum_{k=1}^{d} I(\hat{\boldsymbol{\theta}}_k)) = O(d)$. Therefore, we have

$$\mathbb{E}\|\hat{\boldsymbol{\theta}}_{\mathrm{KL}-W} - \boldsymbol{\theta}_{\mathrm{KL}}^*\|^2 = O(\frac{d}{d^2 n^2}) = O(\frac{1}{dn^2}).$$

We know $\mathbb{E}\|\boldsymbol{\theta}_{\mathrm{KL}}^* - \boldsymbol{\theta}^*\|^2 = O(N^{-1})$ from Liu and Ihler (2014). Then we have

$$\mathbb{E}\|\hat{\boldsymbol{\theta}}_{\mathrm{KL}-W} - \boldsymbol{\theta}^*\|^2 \approx \mathbb{E}\|\hat{\boldsymbol{\theta}}_{\mathrm{KL}-W} - \boldsymbol{\theta}_{\mathrm{KL}}^*\|^2 + \mathbb{E}\|\boldsymbol{\theta}_{\mathrm{KL}}^* - \boldsymbol{\theta}^*\|^2 = O(N^{-1} + (dn^2)^{-1}).$$

The proof of this theorem is complete. $\square$