[Reviews · NeurIPS 2016]

Reviewer 1

Summary

This paper is concerned with one-shot aggregation local models learnt through maximum likelihood estimation on different data repositories in a privacy-preserving setting. The typical approach is to just do linear averaging of the model parameters. That provides nearly optimal asymptotical error rates but can fail in practice for e.g. latent variable models. A more elaborate approach finds a single “centroid” model that minimises the sum of KL-divergences to the individual models. This is hard in practice due to the difficulty of properly estimating KL divergences. The authors propose how to reduce the variance of the KL average distance by leveraging control variates and weightings of the bootstrap samples.

Qualitative Assessment

The paper is well written and the experiments sensible although probably not extensive enough. I'd recommend leaving out the verification of the convergence rates (it can be supplementary material) and adding more performance evaluation. I'm not an expert in this subject so can't accurately judge the relevance of the work, but the application feels a bit niche as it improves the efficiency of a specific existing one-shot averaging method. The authors should make a bigger case for the importance of one-shot averaging in contrast with communication efficient methods such as in Jordan's et al. Communication-efficient distributed statistical learning. Nits: Use \left and \right for the parenthesis in Theorem 2 Algorithm 1, line 5: Fish -> Fisher

Confidence in this Review

2-Confident (read it all; understood it all reasonably well)


Reviewer 2

Summary

A very well written paper on an important topic in the Hadoop/Spark/... era. The authors address the problem of parallel maximum likelihood estimation. Motivated by the drawbacks of naive parameter averaging, they call upon KL-averaging as a remedy. While KL-averaging is not a new idea, previously proposed algorithms are inefficient, making the beautiful theory, simply inapplicable. By successfully designing two, Bootstrap flavoured, practical algorithms, the theory is finally applicable to bigdata scale.

Qualitative Assessment

The authors successfully import ideas from the classical statistical literature, to cutting edge distributed machine learning algorithms. I find that their contribution is not only in the solutions offered in the manuscript, but by uncovering this statistical literature to the machine learning community. Some language errors still remain in current version of the manuscript.

Confidence in this Review

3-Expert (read the paper in detail, know the area, quite certain of my opinion)


Reviewer 3

Summary

The paper discusses the problem of aggregating many models trained on separate data sets to produce one global model; the problem is equivalent to one-shot distributed learning. The authors explain why linear averaging of model parameters is inadequate, and propose two new estimators. They provide compelling theoretical and practical evidence in favor of their proposal.

Qualitative Assessment

The authors provide a good summary of their problem and related approaches, and justify their approach well. I am convinced that they are solving a generic problem with a simple, effective method. Questions: * Why do the theorems (!) on linear model averaging fail in your numerical examples? Are there extra restrictions on the model (eg, strong convexity) required by the theorems in Zhang et al or in Rosenblatt and Nadler? Comments: * Grammar could be improved. Many small conjugation errors. Occasionally "Fish information" is used in place of "Fisher information". * Line 145 is unclear. Why does this constitute "zero variance"? * Line 241: real *world* datasets

Confidence in this Review

2-Confident (read it all; understood it all reasonably well)


Reviewer 4

Summary

The paper considers two variance reduction methods for bootstrap sampling. The motivation is when one obtains different models, e.g., with a resampling method, and the models are on different machines. A straightforward version has drawbacks like linear averaging methods, and a version based on a weighted M estimator is proposed that has the advantages of KL-averaging. Given that a motivation is that existing methods such as linear averaging are nearly optimal in terms of asymptotic error rates but less than ideal in practice, it would be good to see a more detailed evaluation/discussion of that. There is a nice discussion of the theory in Sec 2, so I'm referring to an empirical evaluation, i.e., illustrate empirically those tradeoffs. That would help to understand the strengths and limitations of the proposed method. The main results are in the form of two variance reduction theorems. The proofs are not in the main text, but the discussion around them is good. The empirical results focus on an evaluation of convergence rates of the theory and average LL error, and they are relatively well done. It is unfortunate, however, that the empirical results focus on a latent variable problem where comparison with the direct linear averaging method is not possible. That is one of the main motivations, and that makes it difficult to evaluate some of the theoretical claims made earlier.

Qualitative Assessment

The paper considers two variance reduction methods for bootstrap sampling. The motivation is when one obtains different models, e.g., with a resampling method, and the models are on different machines. A straightforward version has drawbacks like linear averaging methods, and a version based on a weighted M estimator is proposed that has the advantages of KL-averaging. Given that a motivation is that existing methods such as linear averaging are nearly optimal in terms of asymptotic error rates but less than ideal in practice, it would be good to see a more detailed evaluation/discussion of that. There is a nice discussion of the theory in Sec 2, so I'm referring to an empirical evaluation, i.e., illustrate empirically those tradeoffs. That would help to understand the strengths and limitations of the proposed method. The main results are in the form of two variance reduction theorems. The proofs are not in the main text, but the discussion around them is good. The empirical results focus on an evaluation of convergence rates of the theory and average LL error, and they are relatively well done. It is unfortunate, however, that the empirical results focus on a latent variable problem where comparison with the direct linear averaging method is not possible. That is one of the main motivations, and that makes it difficult to evaluate some of the theoretical claims made earlier.

Confidence in this Review

2-Confident (read it all; understood it all reasonably well)


Reviewer 5

Summary

This article proposes two new distributed algorithms to ameliorate the original issue with the bootstrapped KL averaging method. The article shows that a naive bootstrapped KL averaging would result in the same computational burden in combining the estimation even the data is first partitioned for local estimation. The main reason is due to the variance contained in the bootstrap sample. The paper then proposed a variance reduction technique to fix this issue. In particular, the algorithm tries to quantify the bootstrap variance by measuring the difference between the local estimator and local bootstrapped estimator, and correct the global KL estimator based on the quantification. The theory shows that by adding this correction, the convergence rate can be improved to O(N^{-1} + (dn^2)^{-1}).

Qualitative Assessment

This is a successive work correcting previous research on using KL averaging combining subset estimators. I think the most appealing point for using KL averaging, despite the computational issue, is its power in dealing with latent variable models. There is another line of work in using geometric median to combine subset estimators the authors might want to compare to, for example, Minsker (2013) and Hsu and Sabato (2013). These algorithms are simple and efficient in most cases, but might not be doing well for latent variable models. The variance reduction technique used in this article is very similar to the de-bias technique used in Javanmard and Montanari (2015) and Lee et a. (2015), so the theoretical contribution is kind of limited. There are two caveats for the proposed approach. The first lies in the resampling. If the original model is complicated in sampling, then the additional resample requirement from the KL averaging could ruin the whole framework. Second, the variance reduction requires to compute the Fishier information matrix and the inverse, which will adds an O(p^3) computational burden. This will also become huge when p is large. ======================== after reading feedback ==================== Previously I thought the advantage of this paper over the geometric median or the simple averaging approach is that it can handle latent variable models, such as estimating mixture of Gaussians. It is well known that the latent variable model has this "label switching" problem (the parameters are equivalent up to some permutation), so a direct averaging of the parameters obtained from different subsets might result in bad outcomes due to the permutation. The KL averaging method solves this problem by avoiding directly averaging the sub-estimators. Instead, it resamples data from each subset and pools the new data together to estimate the final result, so as to avoid the "label switching" issue. However, as now I double checked the paper, I found that the variance reduction technique introduced in this paper actually ruined this nice property of the original KL averaging. If you take a look at (7), which is the main formula in this article, you will realize that (7) also suffers from the "label switching" problem if \theta's are up to some permutation. Then we arrive at some tricky scenario. On the one hand, the KL averaging can handle the latent variable model, but the efficiency is bad (according to this article) compared to the direct averaging. On the other hand, the enhanced KL averaging with variance reduction can improve the efficiency to the same level as direct averaging, but loses the ability to hand latent variable models (essentially the same as direct averaging). So I kind of doubt how much can be gained practically by using this method, which involves much heavier computation compared to averaging. It is a pity there is only one round of author feedback and it is my fault not to pose this question to the author in the previous round. I would love to know how the authors address this problem. But given this is my problem not to ask, I will raise my score of technical quality from 2 --> 3

Confidence in this Review

2-Confident (read it all; understood it all reasonably well)